# Small-sample learning reveals propionylation in determining global protein homeostasis

Ke Shui [1,6], Chenwei Wang [1,6], Xuedi Zhang [2], Shanshan Ma[1], Qinyu Li[1], Wanshan Ning [1], Weizhi Zhang [1], Miaomiao Chen[1], Di Peng [1], Hui Hu [1], Zheng Fang [3], Anyuan Guo [1], Guanjun Gao [2], Mingliang Ye [3], Luoying Zhang [1,4] ✉ & Yu Xue [1,5] ✉

Proteostasis is fundamental for maintaining organismal health. However, the mechanisms underlying its dynamic regulation and how its disruptions lead to diseases are largely unclear. Here, we conduct in-depth propionylomic profiling in *Drosophila*, and develop a small-sample learning framework to prioritize the propionylation at lysine 17 of H2B (H2BK17pr) to be functionally important. Mutating H2BK17 which eliminates propionylation leads to elevated total protein level in vivo. Further analyses reveal that H2BK17pr modulates the expression of 14.7–16.3% of genes in the proteostasis network, and determines global protein level by regulating the expression of genes involved in the ubiquitin-proteasome system. In addition, H2BK17pr exhibits daily oscillation, mediating the influences of feeding/fasting cycles to drive rhythmic expression of proteasomal genes. Our study not only reveals a role of lysine propionylation in regulating proteostasis, but also implements a generally applicable method which can be extended to other issues with little prior knowledge.

In eukaryotic cells, protein homeostasis, or proteostasis, is an essential regulatory mechanism to precisely balance a functional proteome[1–4]. It has been estimated that over 2000 human proteins, including ~400 protein synthesis factors, ~300 molecular chaperones, and >1000 regulators in ubiquitin-proteasome system (UPS) and autophagy-lysosomal system (ALS), are involved in orchestrating proteostasis[1,3–8]. These proteins form a highly complex proteostasis network (PN), which temporally and spatially regulates protein synthesis, folding, conformation, stability, trafficking, aggregation and degradation to ultimately determine the composition, abundance and activity of the proteome[3,4]. Proteostasis can be significantly affected by physiological, metabolic, and environmental stimuli, and its capacity and plasticity declines during ageing and under disease conditions[2,9]. In particular, dysregulation of proteostasis is highly associated with diseases, such as neurodegenerative diseases, metabolic disorders, cardiovascular diseases and cancer[1,7,10,11].

Post-translational modifications (PTMs), such as phosphorylation and lysine acylation, contribute to the dynamic regulation of proteostasis[2,4,11,12]. Lysine acylation comprises a diverse class of evolutionarily conserved PTMs that are directly linked to cellular metabolism[13]. Recent studies have demonstrated that the best-known acylation, lysine acetylation (Kac), is involved in proteostasis by

[1]Key Laboratory of Molecular Biophysics of Ministry of Education, Hubei Bioinformatics and Molecular Imaging Key Laboratory, Center for Artificial Intelligence Biology, College of Life Science and Technology, Huazhong University of Science and Technology, Wuhan 430074 Hubei, China. [2]School of Life Science and Technology, ShanghaiTech University, 393 Middle Huaxia Road, 201210 Shanghai, China. [3]CAS Key Laboratory of Separation Science for Analytical Chemistry, Dalian Institute of Chemical Physics, Chinese Academy of Sciences, 116023 Dalian, China. [4]Hubei Province Key Laboratory of Oral and Maxillofacial Development and Regeneration, Wuhan 430022 Hubei, China. [5]Nanjing University Institute of Artificial Intelligence Biomedicine, Nanjing 210031 Jiangsu, China. [6]These authors contributed equally: Ke Shui, Chenwei Wang. ✉e-mail: zhangluoying@hust.edu.cn; xueyu@hust.edu.cn

regulating gene expression of key transcription factors (TFs) and molecular chaperones, mainly through epigenetic modifications of histone proteins[2,14]. As a less studied acylation, lysine propionylation (Kpr, three carbon molecules) was first discovered in 2007 as a histone modification, and later it was found on non-histone proteins as well[15,16]. Kpr exhibits functional difference from Kac in vitro, but its physiological role is largely unknown[17]. Moreover, propionyl-CoA is a key intermediate of amino acid, fatty acid and cholesterol catabolism, linking Kpr to metabolic processes distinct from Kac[18]. Human individuals carrying mutations in genes encoding propionyl-CoA carboxylase or methylmalonyl-CoA mutase are affected with propionic acidemia (PA) and methylmalonic acidemia (MMA), which are characterized by life-threatening acute metabolic decompensation (AMD) with excessive protein catabolism[19]. These enzymes are responsible for converting propionyl-CoA to succinyl-CoA, and thus their deficiency results in accumulation of propionyl-CoA which in turn leads to increased protein propionylation[20]. However, it is not known whether Kpr is involved in regulating proteostasis.

To investigate the function of Kpr, we conduct propionylomic profiling of *Drosophila* heads and quantify 171 propionylated proteins containing 344 Kpr sites. Due to the varying functions of PTM sites[21,22], we hypothesize that only a small subset of Kpr sites might be functionally important in regulating biological processes. To avoid over-fitting during model training, we develop a small-sample learning framework, prediction of Kpr sites with functional relevance (KprFunc). KprFunc scores 29 Kpr sites to be functionally important and we successfully validate one of the top hits, propionylation at lysine 17 of histone H2B (H2BK17pr). Remarkably, this site is evolutionarily conserved and we find that the orthologous human H2B site (H2BK23) is propionylated as well. Mutating this site to alanine (H2BK17A) which eliminates propionylation leads to increased total protein level in flies, whereas increasing H2BK17pr is accompanied by reduced total protein level in fly and human cells. Further analyses reveal that H2BK17pr modulates the expression of 14.7–16.3% of PN genes, and determines global protein level by regulating the expression of UPS genes. To explore the dynamic regulation of H2BK17pr, we find that H2BK17pr is rhythmically regulated by daily feeding/fasting cycles and in turn drives rhythmic expression of proteasomal genes. Lastly, we characterize enzymes that modify H2BK17pr and re-constructed a signaling network based on our multi-omic data, which predicts how H2BK17pr modulates the expression of PN genes. Taken together, our study not only uncovers a role for H2BK17pr in regulating proteostasis via the UPS system, but also provides an explanation for the excessive protein catabolism occurring in PA and MMA, which have remained mysteries thus far. In addition, this is the first successful application of small-sample learning strategy in artificial intelligence (AI) for scientific research, and we anticipate that the framework we developed here can be adapted to resolve many other issues with little prior knowledge.

## Results

### Dynamic Kpr profiling in fly heads

To investigate the physiological function of Kpr, we sought to identify proteins and lysine sites that are propionylated. We collected wild-type (WT) fly heads at four time points throughout the day to ensure that we capture the temporal dynamics of Kpr (Fig. 1a)[23,24]. In addition, we collected the heads of flies mutant for the core circadian clock gene *period* (*per⁰*) at two different time points[25]. For each sample, whole-cell lysates were digested by trypsin and labeled with one of the tandem mass tag (TMT) 6-plex reagents. Propionylated peptides were then isolated by immuno-affinity enrichment using the pan-propionylation antibody. The 6 samples were mixed with equal amounts and analyzed by liquid chromatography-tandem mass spectrometry (LC-MS/MS).

Altogether, we identified 330 propionylated peptides from the 6 samples, the majority of which (147; 44.55%) had ≥2 spectral counts, indicating a promising quality of propionylomic profiling at the peptide level (Supplementary Fig. 1a and b). We next mapped these peptides to their corresponding protein sequences and obtained 344 Kpr sites in 171 proteins (Supplementary Data 1a). For each Kpr site, MaxQuant[26] computationally assigned a localization probability (LP) score, which ranges from 0 to 1 and a higher LP score denotes a higher probability for a site to be a true Kpr site. In our data, up to 336 (97.7%) Kpr sites have a LP score of 1, indicating a high reliability of Kpr site identification, and we detected only one Kpr site for the majority of the proteins (100; 58.48%) (Fig. 1b and Supplementary Fig. 1c)[26]. Since the number of Kpr sites detected here is considerably fewer than the 2023 Kac sites reported in flies, we wondered if this could be due to a lower abundance of propionyl-CoA relative to acetyl-CoA[27,28]. We measured the level of these two co-enzymes in fly heads but observed no significant difference (Fig. 1c). This implies that the complexity of Kpr in proteins may be comparable to Kac.

We further analyzed the amino acid preferences surrounding Kpr sites, and found that glycine was over-represented at +2 position while lysines were enriched at +7 and +8 positions downstream of the Kpr site (Supplementary Fig. 1d). An enrichment analysis of Gene Ontology (GO) biological processes revealed that the propionylated proteins are markedly enriched in mitochondria and nucleosome-related pathways (Fig. 1d). To identify Kpr sites that are temporally regulated, we calculated the largest fold change (FC) of propionylation level at each site by dividing the maximum intensity at the peak time point by the minimum intensity at the trough time point (Supplementary Data 1b). There are 79 (~23%) lysine sites that exhibit a >1.5 FC of propionylation level throughout the day, including H2BK17pr (FC of 2.0), K249pr of mitochondrial magnesium exporter 1 (MME1, 1.9 FC), and K362pr of Mitochondrial trifunctional protein α subunit (MTPα, 1.9 FC) (Fig. 1e; Supplementary Data 1b). Among the 79 Kpr sites that show prominent temporal variation, propionylation level at 59 (~75%) sites are affected by mutation of clock gene *per* (*per⁰*) with >1.2 FC (Supplementary Fig. 1e and 1f, Supplementary Data 1b).

To evaluate the specificity of our Kpr enrichment process, MaxQuant[26] was re-used to simultaneously search both Kpr and Kac peptides. This resulted in the identification of 114 non-modified and 488 modified peptides, while the majority (363/488) of modified peptides are propionylated peptides (Supplementary Fig. 1g; Supplementary Data 1c and e). Also, we manually checked the MS/MS spectra of propionylated peptides for the diagnostic ion at $m/z$ 140.1075 for Kpr[29], using a program developed in house and adapted from Glyco-Decipher, a tool for analysis of glycopeptide spectra[30]. This fragment was observed in all propionylated peptides detected (Supplementary Fig. 2).

### Small-sample learning predicts the functional relevance of Kpr sites

Due to weak constraints during evolution, PTM sites display differential functions and the extent of their influences on biological processes vary[21,22]. Previously, it was estimated that ~65% of the phosphoproteome is nonfunctional[21], and only ~10% of phosphorylation sites (p-sites) might be functionally important[22]. Thus, we hypothesized that only a small proportion of the 344 identified Kpr sites are be functionally important. However, only 13 Kpr sites have been reported to be functional based on the literature (Supplementary Data 2a), and directly training a model from such a small-size data will be highly biased and easily lead to over-fitting.

To predict the functional relevance of Kpr sites, we developed a small-sample learning framework called KprFunc with three steps, including sequence embedding, pre-training, and fine-tuning (Fig. 2a). First, we integrated our propionylomic data with previously reported Kpr sites[27,31], and compiled a benchmark data set containing 1707 known and non-homologous Kpr sites in 890 proteins from both eukaryotes and prokaryotes (Supplementary Data 2b). Previously, we developed a Group-based Prediction System 5.0 (GPS 5.0) algorithm,

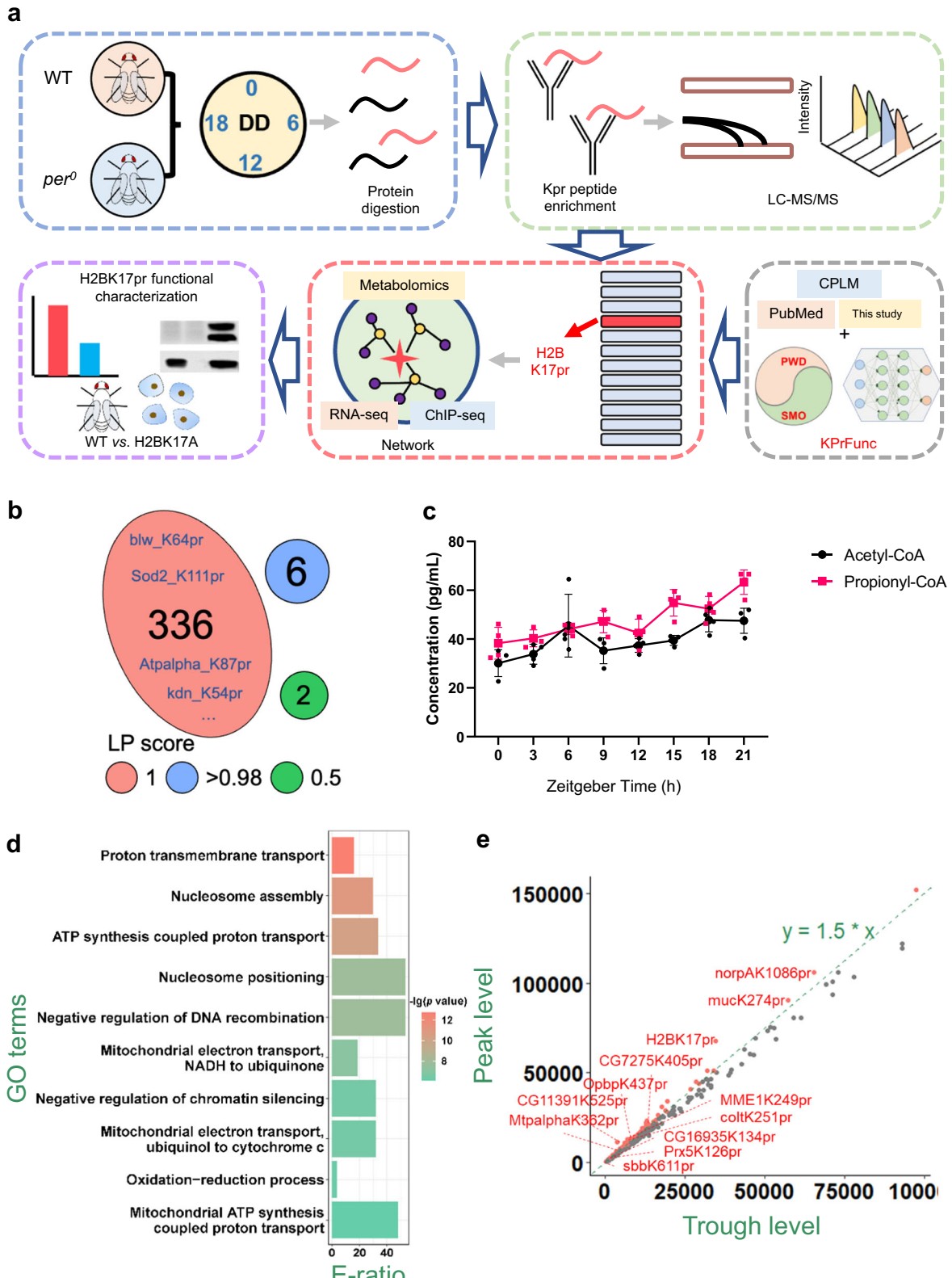

in which two methods including position weight determination (PWD) and scoring matrix optimization (SMO) were designed for prediction of kinase-specific p-sites[32]. Here PWD and SMO were adopted to embed the short sequences around Kpr sites into one dimensional (1D) vectors, which were iteratively optimized using the penalized logistic regression (PLR) algorithm. To learn the sequence features of Kpr sites, an initial model, KprFunc-i, was pre-trained by a 4-layer deep neural network (DNN) framework, using the optimal 1D vectors as the input (Fig. 2a; Supplementary Data 2c). To further learn the functional characteristics of Kpr sites, the KprFunc-i model was fine-tuned with the 13 Kpr sites known to be functional, using a cutting-edge small-sample learning method of Model-Agnostic Meta-Learning (MAML)[33].

For prediction of Kpr sites, we performed the tenfold cross-validation to evaluate the performance. KprFunc-i achieved an area

**Fig. 1 | Dynamic propionylomic profiling of fly heads. a** Workflow of this study. In brief, WT and $per^0$ fly heads were collected at Circadian Time (CT) 0, 6, 12 and 18 (CT0 is defined as the start of the subjective day) on the first day of constant darkness (DD1), while $per^0$ fly heads were collected at CT6 and 18 on DD1. LC-MS/MS-based propionylomic was conducted. An AI-based tool KprFunc was implemented for predicting functional Kpr sites. H2BK17pr was predicted to be highly functional and also showed prominent temporal variation, and thus was selected for further functional characterization using multi-omics approach and molecular analysis in flies and mammalian cells. **b** The LP scores of Kpr sites. **c** Plot shows quantification of acetyl-CoA and propionyl-CoA level in WT fly heads using enzyme-linked immunosorbent assay (ELISA) under time-restricted feeding during ZT0 to ZT12 ($n = 4$ biologically independent experiments, two-tailed Mann–Whitney $U$ test for unpaired comparisons). Data are presented as the mean ± SD. **d** GO-based enrichment analysis of propionylated proteins quantified in this study. E-ratio, enrichment ratio (one-sided hypergeometric test). **e** The peak and trough propionylation levels of Kpr sites. A number of sites with propionylation level FC > 1.5 are labeled in red. Source data are provided as a Source Data file.

---

under the curve (AUC) value of 0.8454, exhibiting a > 21.3% increase of accuracy than other methods including the original GPS 5.0 method, DNN only, support vector machines (SVMs), random forest (RF) and $K$-nearest neighboring (KNN) (Fig. 2b). For KprFunc-i, an analysis of t-distributed stochastic neighbor embedding (t-SNE) demonstrated that Kpr and non-Kpr sites can be distinguished (Supplementary Fig. 3a). For predicting the functional relevance of Kpr sites, the 5-fold cross-validation was performed due to data limitation. KprFunc-i only got an AUC value of 0.3674, whereas the fine-tuned model of KprFunc reached a highly promising AUC value of 0.8852 (Fig. 2c). Again, the t-SNE results supported that KprFunc can distinguish functional Kpr sites from other Kpr sites (Supplementary Fig. 3b). Under a threshold of specificity (Sp) ≥ 90% (Supplementary Data 2d), the confusion matrices demonstrate the promising performance of KprFunc-i in prediction of Kpr sites and KprFunc in prediction of functional Kpr sites, respectively (Fig. 2d and e). The distribution of scores predicted by KprFunc-i and KprFunc are shown for all 13 known functional Kpr sites (Supplementary Fig. 3c and d). Based on these results, the 13 Kpr sites known to be functional in general received higher scores assigned by KprFunc but not KprFunc-i. This means the MAML-based refinement markedly enhanced the prediction of functionality.

To demonstrate the superiority of pre-training followed by fine-tuning in KprFunc, the 13 Kpr sites known to be functional were directly used for model training. Using the fivefold cross-validation, AUC values of DNN, SVMs, RF, and KNN algorithm ranged from 0.7607 to 0.9768 (Supplementary Fig. 3e), implying that the functional characteristics of Kpr sites were well captured. However, when using the 1707 known Kpr sites for testing, KprFunc showed a promising accuracy while other models could not distinguish Kpr sites from non-Kpr sites (Supplementary Fig. 3f), indicating that these ab initio models were highly biased and failed to learn the sequence features of Kpr sites. To further demonstrate the superiority of KprFunc on prediction of functional Kpr sites, 165 of 620 (26.61%) fly proteins previously identified to be circadianly regulated in abundance were predicted to contain at least one functional Kpr site (Supplementary Data 2e), implying that a considerable proportion of rhythmic proteins are influenced by propionylation[34]. GO-based enrichment analysis indicates that these Kpr-regulated proteins are enriched in transport and chromatin/transcription-related processes, implicating a role for circadian propionylation in modulating these pathways (Fig. 2f). Additional details on KprFunc are elaborated in Supplementary Discussion.

For convenience, an online service of KprFunc was developed (http://kprfunc.biocuckoo.cn/) for prediction of either general or functional Kpr sites. Single or multiple protein sequences in FASTA or eukaryotic linear motif (ELM)[35] format can be submitted (Supplementary Fig. 3g). The prediction results will be shown in a tabular list, including protein accession number/ID, propionylation position, residue type, predictor used, flanking peptide, predicted score, and predefined cut-off value (Supplementary Fig. 3h).

### H2BK17pr is a conserved Kpr site driven by daytime TRF

Using KprFunc, we predicted 29 Kpr sites from our propionylomic profiling to be functionally important (Supplementary Data 2f). Notably, one of the top sites predicted to be functional is H2BK17pr, which is also one of the sites that show markedly robust temporal variation

(Fig. 1a and d; Supplementary Data 1b and 2e). H2BK17pr is located at the N-terminal flexible tail (Fig. 3a). In eukaryotes, H2B is not encoded by a single gene and there are multiple variants. From a public database HistoneDB 2.0[36], we obtained 7,671 H2B protein sequences in eukaryotes. An evolutionary analysis demonstrated that these H2B genes could be classified into 6 families, including H2B.1, canonical H2B, sperm H2B, H2B.Z, subH2B and H2B.W, whereas all the 494 *Drosophila* H2B genes belong to the canonical H2B family (Fig. 3b). By multiple sequence alignment of all H2B genes, variants containing the K17 site were identified (Supplementary Fig. 4). This site is present in 4496 (58.61%) of the H2B variants, mostly in canonical H2B (56.99%) and H2B.1 (100.00%) families (Fig. 3c). Moreover, we found this fly H2BK17 site to be evolutionarily conserved in a considerable proportion of H2B variants in mouse and human, further supporting the functionality prediction of KprFunc (Fig. 3d).

To validate that the H2BK17 site in *Drosophila* is indeed propionylated, we generated an antibody that specifically detects this particular modification but not acetylated or non-modified peptide (Supplementary Fig. 5a and b). We were able to detect H2BK17pr signal in WT fly heads, while a fly line carrying H2BK17A lysine mutation which eliminates propionylation at this site does not exhibit H2BK17pr signal (Fig. 3e and Supplementary Fig. 5c). Moreover, propionate treatment could increase H2BK17pr in a dose-dependent manner in fly S2 cells (Fig. 3f). Given the conservedness of the H2BK17 site, we next tested whether the H2BK17 equivalent site (H2BK23) is propionylated in the mammalian system. We detected H2B propionylation in mouse NIH 3T3 cells, human U2OS cells and Human Embryonic Kidney 293 (HEK293) cells using the H2BK17pr antibody, and the propionylation signals are enhanced by propionate treatment compared to acetate treatment (Fig. 3g and h). In addition, we used the H2BK17pr antibody to immunoprecipitate H2B in *Drosophila* heads and HEK293 cell extracts, respectively, and LC-MS/MS analysis was conducted to validate the propionylated peptides. To ensure data quality, peptide-spectrum matches (PSMs) were manually annotated[30]. We found that all $y$-ions could be correctly matched and at least three pairs of $b$- and $y$-ions could be unambiguously pinpointed, while residual non-fragmented precursor ions were also detected (Fig. 3i and j). Besides H2BK17pr containing peptides, we noticed the presence of peptides bearing H2BK17ac along with some other H2B peptides (Supplementary Fig. 5d and e). Nonetheless, these results clearly demonstrate the presence of propionylation at H2BK23 in HEK293 cells.

Rhythmic feeding events regulate daily oscillations in various cellular and physiological processes[37,38]. Given that propionyl-CoA is a key intermediate of amino acid, fatty acid and cholesterol catabolism, we investigated whether the daily variation of H2BK17pr is modulated by rhythmic feeding[18]. We subjected flies to a time-restricted feeding (TRF) paradigm. When feeding is restricted to the day which is when flies usually eat, the oscillation of H2BK17pr is enhanced compared to *ad libitum* (AL) feeding (Supplementary Fig. 6a–c)[39]. On the other hand, restricting feeding to the night eliminates the oscillation of H2BK17pr. Moreover, the rhythmic regulation of feeding cycles on H2BK17pr depends on the clock, as $per^0$ flies do not exhibit cycling of H2BK17pr under daytime restricted feeding condition (Supplementary Fig. 6d and 6e). Based on these findings, we propose that feeding cycles generate rhythmic signals that lead to cyclic propionylation/

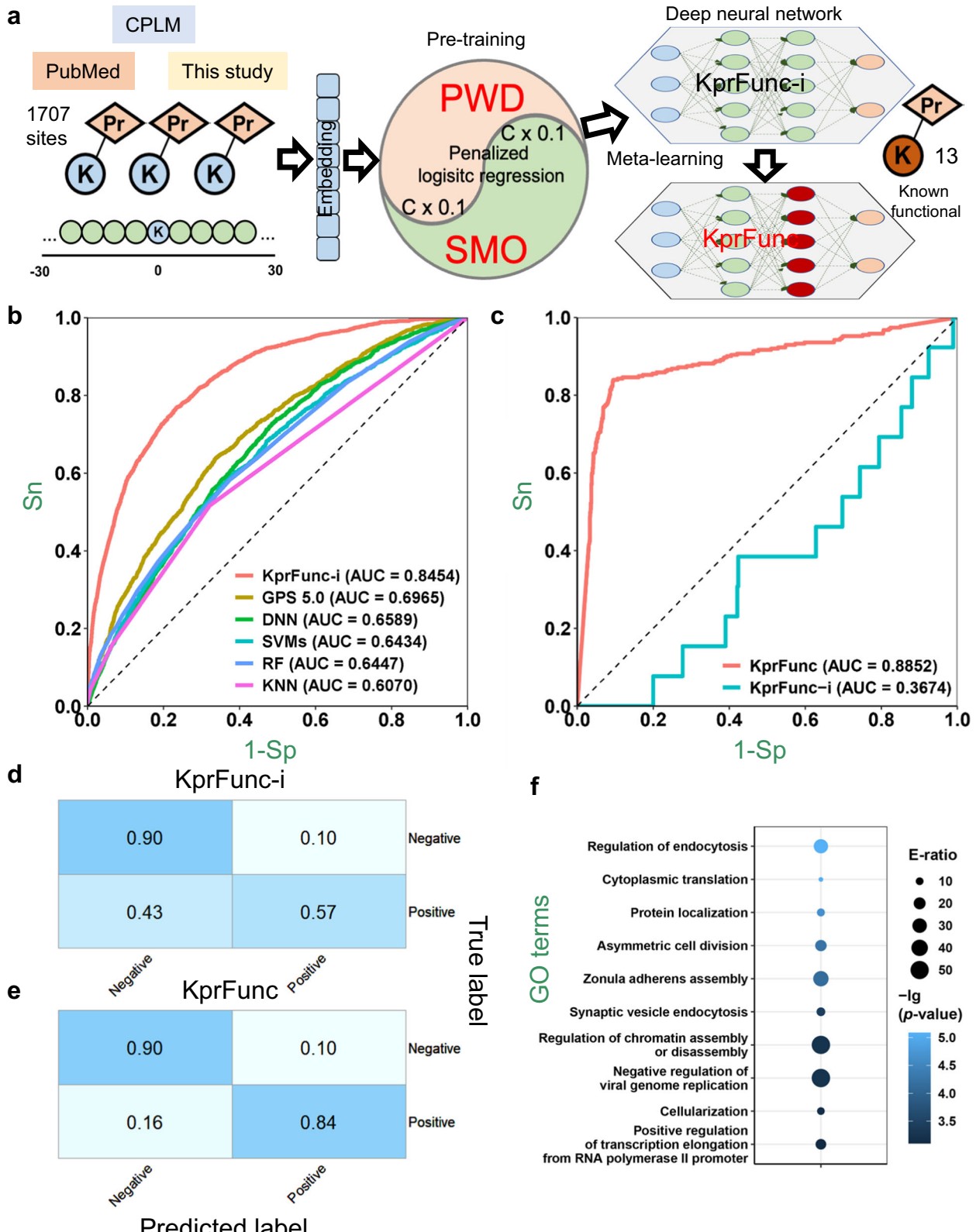

**Fig. 2 | The small-sample learning framework and performance of KprFunc.**
**a** The architecture of KprFunc, which incorporates PWD, SMO, PLR, DNN and
MAML algorithms. **b** ROC curves and AUC values of tenfold cross-validations of
KprFunc-i and other methods. **c** ROC curves and AUC values demonstrating the
performance of KprFunc-i and KprFunc in distinguishing functional Kpr sites from
other sites. **d** The confusion matrix of KprFunc-i under the medium threshold
(Sp ≥ 90%). **e** The confusion matrix of KprFunc under the medium threshold
(Sp ≥ 90%). **f** GO-based enrichment analysis of circadian proteins with functional
Kpr sites predicted by KprFunc. E-ratio, enrichment ratio (one-sided
hypergeometric test).

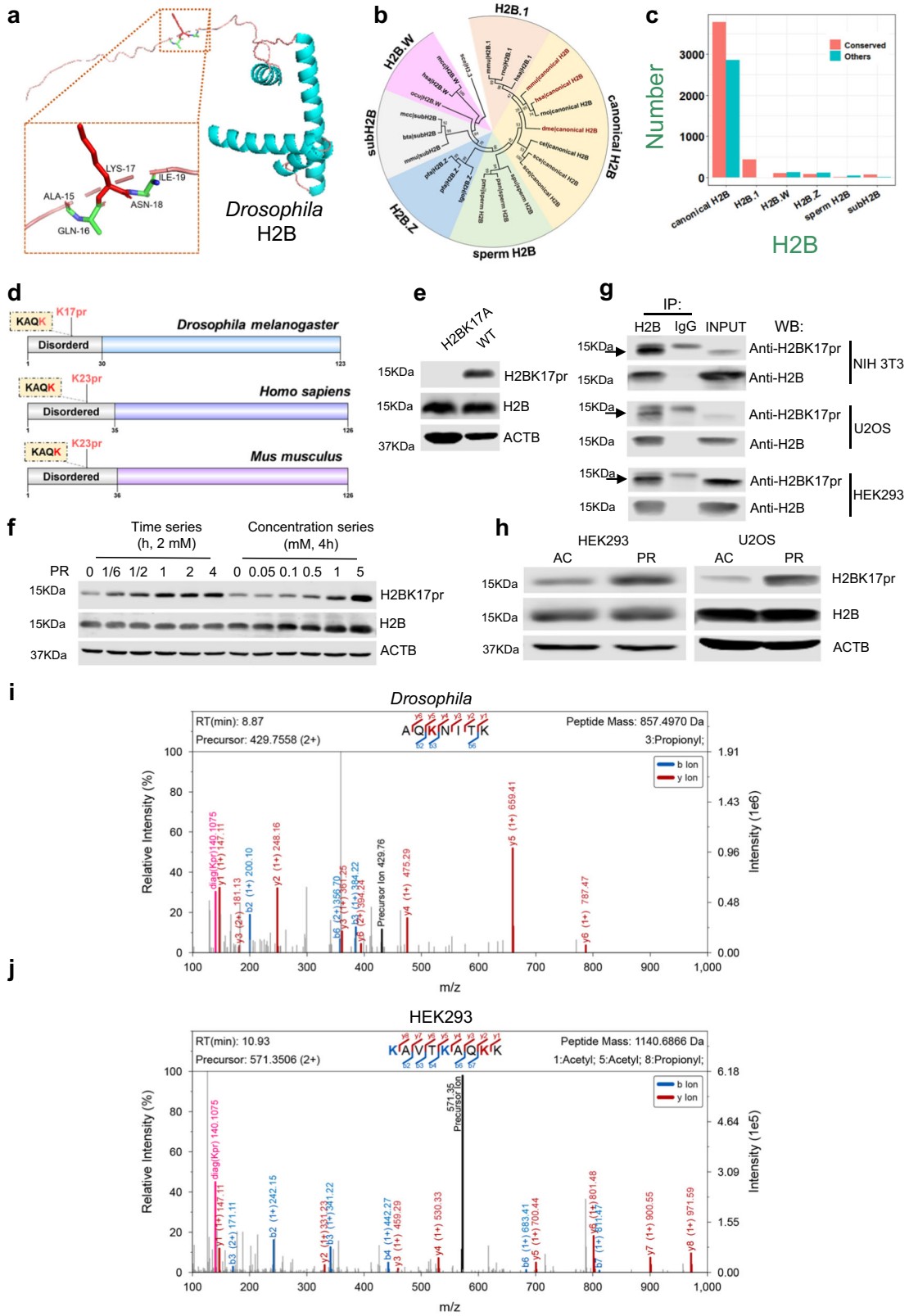

depropionylation at H2BK17, while this process may be gated by the circadian clock.

## H2BK17pr is critical for maintaining global proteostasis

To investigate the physiological function of H2BK17pr, we first monitored the circadian rhythm of WT and H2BK17A flies because this modification displayed circadian oscillation. Locomotor rhythm is not affected by H2BK17A, and none of the core circadian clock genes are significantly altered either (Supplementary Fig. 7). We next assessed various physiological indices and observed significantly increased total protein level in H2BK17A flies compared to WT, indicating that H2BK17A alters proteostasis (Fig. 4a). Weight, glucose and glyceride levels do not show substantial difference between the mutants and the controls (Fig. 4b–d). Given that H2BK17pr oscillation is regulated by

**Fig. 3 | The structural and evolutionary analysis of H2BK17pr. a** The protein structure of *Drosophila* H2B predicted by AlphaFold. **b** The evolutionary tree of H2B proteins from various species and different families. **c** The number of H2B variants with (conserved) or without (others) the K17 site or its homologous site. **d** Amino acid sequence KAQK is present in H2Bs from *Drosophila melanogaster*, *Homo sapiens* and *Mus musculus*, with the latter K (labeled in red) being potentially propionylated. **e** Western blots of proteins from whole-head extracts of WT and H2BK17A flies probed with indicated antibodies. β-actin (ACTB) is used as a loading control. **f** Western blot of protein extracts from S2 cells treated with sodium propionate (PR) at indicated concentrations and time durations. ACTB is used as a loading control. **g** Immunoprecipitation of protein extracts from NIH3T3, U2OS

and HEK293 cells using anti-H2B or IgG control. The precipitates are blotted with the indicated antibodies. IP immunoprecipitation, WB Western blot. **h** Western blots of protein extracts from HEK293 and U2OS cells treated with sodium propionate and sodium acetate for 24 h at 5 mM. ACTB is used as a loading control. AC sodium acetate, PR sodium propionate. **i** The MaxQuant MS/MS spectra of AQK(pr) NITK peptide identified from *Drosophila* heads. The propionylated K corresponds to K17. **j** The MaxQuant MS/MS spectra of K(ac)AVTK(ac)AQK(pr)K peptide identified from HEK293 cell. The propionylated K corresponds to K23. The blotting experiments were conducted with three independent repeats with similar results (**e**, **g**, **f** and **h**). The values of *m/z* in spectra and source data are provided as a Source Data file.

TRF, we also subjected these flies to daytime restricted-feeding. We observed an increase of total protein level in H2BK17A flies relative to WT, similar to AL condition (Fig. 4a). Because the major source of protein in the fly food comes from yeast, we examined the effects of H2BK17A on yeast intake[40]. We found that H2BK17A flies exhibit significantly reduced yeast intake relative to WT, while as a control, sugar intake was not altered (Fig. 4e and f). This decrease of protein intake in the mutants is probably the consequence of increased total protein level, further corroborating the notion that H2BK17A alters proteostasis. To validate that the increased total protein level in H2BK17A flies is caused by deficient propionylation at this site, we treated S2 cells with propionate which can increase H2BK17pr and observed dose-dependent reduction of total protein level compared to acetate and PBS treatment (Fig. 4g and h). As a control, we also examined the effects of propionate treatment on histone acetylation and observed no substantial alteration of H3ac and H3K27ac (Supplementary Fig. 8a).

The potential effect of H2BK17pr on protein level is reminiscent of the protein catabolism symptom in PA and MMA patients. Therefore, we suspected H2BK23pr, the mammalian equivalent of H2BK17pr, may also influence proteostasis in the mammalian system. In order to test this, we treated several human cell lines with propionate and examined the effects on total protein level (Fig. 4i and j). We found that propionate reduces total protein level in human cardiomyocyte AC16 cells, but not in U2OS cells or HEK293 cells. Consistent with this, propionate treatment elevated anti-H2BK17pr signal which likely reflects increased H2BK23pr level (Fig. 4k). On the other hand, propionate does not alter H2BK23ac level in these cells (Supplementary Fig. 8b–d). Taken together, these findings implicate a conserved role for H2B propionylation in modulating proteostasis, from fly to human.

## H2BK17pr regulates proteostasis via the UPS
To investigate the mechanism by which H2BK17pr regulates proteostasis, WT and H2BK17A flies were subjected to AL or daytime restricted feeding condition. Fly heads were collected every 3 h and then transcriptomic profiling was conducted by RNA sequencing (RNA-seq) (Fig. 1a), with $1.92 \times 10^7$ to $2.60 \times 10^7$ clean reads and 11,572–13,044 mappable genes in all 32 samples (Supplementary Fig. 9a; Supplementary Data 3a). After calculating Fragments per Kilobase of exon per Million fragments mapped (FPKM) values of genes mapped (Supplementary Fig. 9b), a 2-way hierarchical clustering was performed for all RNA-seq data (Supplementary Fig. 9c). Samples of different genotypes and treatment conditions were clearly separated, indicating a high quality of our RNA-seq profiling. By comparison, we identified 2458 and 2715 differentially expressed genes (DEGs) in H2BK17A vs. WT flies under AL or TRF condition, respectively. A GO-based enrichment analysis demonstrated that cytoplasmic translation, heterophilic cell-cell adhesion via plasma membrane cell adhesion molecules, motor neuron axon guidance and neuropeptide signaling pathway were affected by H2BK17A under both conditions (Supplementary Fig. 9d).

Differential expression of translation-related genes in H2BK17A flies are consistent with the altered proteostasis observed. We further examined the effects of H2BK17A on genes that function in the fly PN. We collected 1558 fly PN genes, including 154 protein synthesis factors,

210 molecular chaperones, 802 UPS enzymes/regulators, and 510 ALS regulators (Supplementary Data 3b). We found that all aspects of the PN were considerably affected by H2BK17A, and in total 14.70% and 16.30% of PN genes were differentially regulated under AL and TRF conditions, respectively (Fig. 5a and b; Supplementary Data 3b). More than half of the (130/229 and 130/254) differentially-expressed PN genes were altered both under AL and TRF, implying a congruent mechanism of H2BK17pr in orchestrating global proteostasis under different physiological conditions (Fig. 5c).

To identify direct targets of H2BK17pr among these PN DEGs, we examined H2BK17pr binding in genomes of AL- and TRF-treated fly heads. Chromatin immunoprecipitation followed by sequencing (ChIP-seq) was conducted using the H2BK17pr antibody (Fig. 1a). In total we obtained $1.23 \times 10^8$ and $1.59 \times 10^8$ clean reads with average numbers of 2552 and 5422 mappable genes from AL- and TRF-treated fly heads, respectively (Supplementary Fig. 10a; Supplementary Data 4a and 4b). A 2-way hierarchical clustering was performed for all ChIP-seq data by pairwisely calculating the Spearman's correlation coefficient (Supplementary Fig. 10b). The principle component analysis (PCA) demonstrated that AL-treated and TRF-treated samples could be unambiguously distinguished, indicating a high reliability of our ChIP-seq profiling (Supplementary Fig. 10c). We observed that H2BK17pr is most highly enriched in the promoter regions close to transcription start site (TSS, −1kb to 1 kb bp) (Fig. 5d). Interestingly, the peak number is dramatically increased by nearly threefold in TRF- (103,659) compared to AL-treated (34,572) flies, implying that H2BK17pr regulates the expression of a larger number of genes in response to TRF. In line with this idea, we found that H2BK17pr binding is closer to TSS under TRF (Fig. 5e). Indeed, H2BK17pr binds to the TSS region of 4169 genes under AL and 7582 genes under TRF, with a large overlap of 4004 genes that are bound by H2BK17pr under both conditions (Fig. 5f; Supplementary Data 4a and 4b). By comparison, we found that TRF increases H2BK17pr binding at PN DEGs (Fig. 5g–i), indicating that more PN genes are directly regulated by H2BK17pr under TRF.

In addition, metabolomic profiling was conducted using TRF-WT and TRF-H2BK17A fly heads collected at ZT9 and ZT21 (Fig. 1a; Supplementary Data 5a and 5b). H2BK17A leads to alterations of metabolites enriched in a number of protein metabolism-related pathways, including protein digestion and absorption, beta-alanine metabolism and biosynthesis of amino acids (Supplementary Fig. 11a; Supplementary Data 5c). Indeed, the levels of various amino acids exhibit significant differences between TRF-WT and TRF-H2BK17A, with methionine being most dramatically different (Supplementary Fig. 11b; Supplementary Data 5d).

To further investigate alterations in which aspects of the PN are crucial for the increased protein level observed in H2BK17A flies, we assessed global ubiquitination level and the ratio of ATG8A-I/ATG8A-II which serves as a marker of autophagic activity[41]. Ubiquitination is significantly reduced in H2BK17A flies, consistent with increased total protein level (Fig. 5j and k). On the other hand, the ratio of ATG8A-I/ATG8A-II is not significantly altered, indicating that H2BK17A does not substantially affect autophagy (Supplementary Fig. 12a and 12b). In addition, we treated S2 cells with inhibitors of the proteasome (MG132),

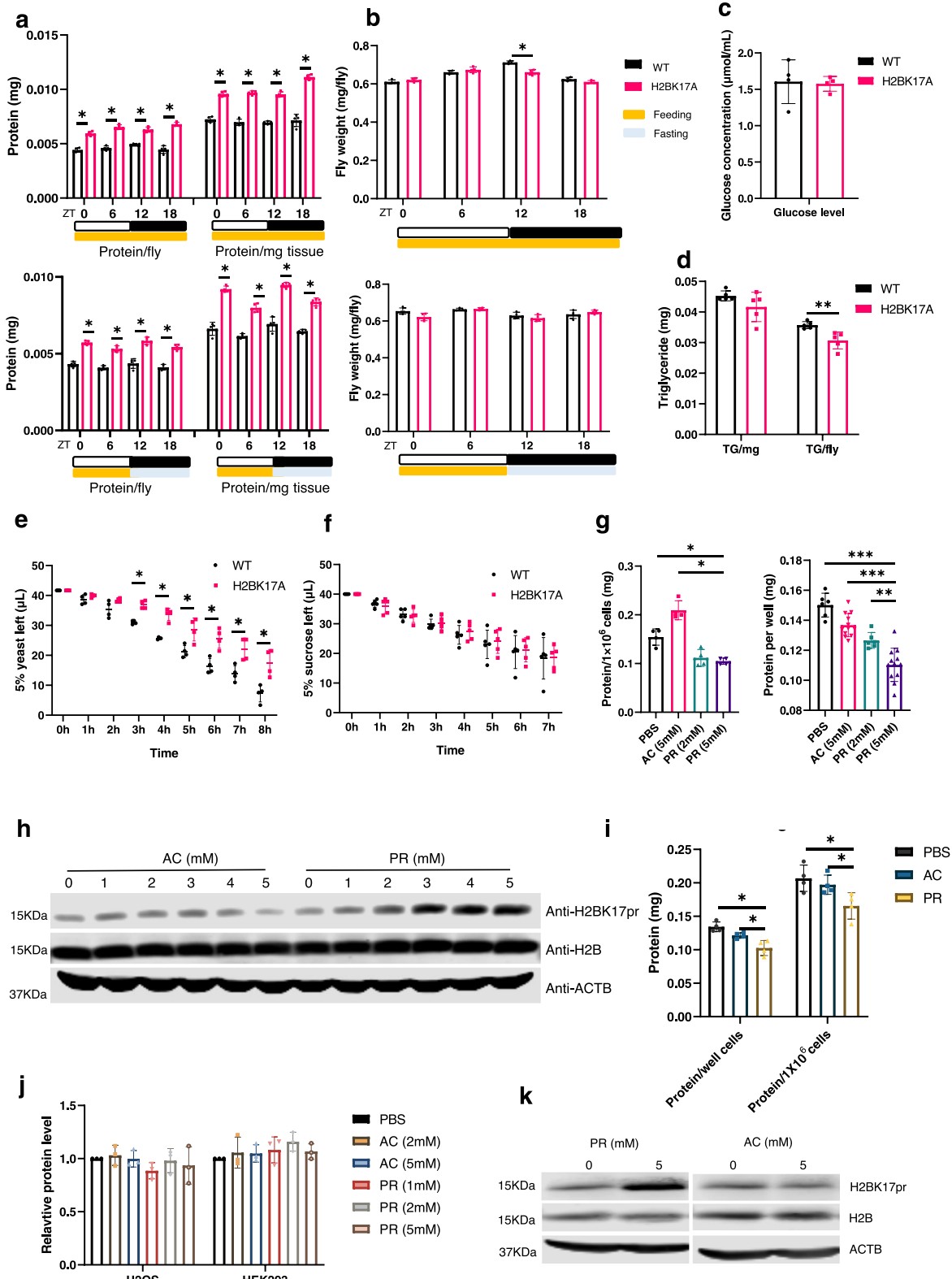

autophagy (chloroquine) and translation (cycloheximide), and found that propionate fails to significantly reduce total protein level only in MG132-treated cells (Fig. 5l, Supplementary Fig. 12c and 12d). Taken together, these results suggest that H2BK17pr modulates total protein level via the UPS, by regulating the expression of UPS genes.

To verify that the UPS DEGs in H2BK17A flies are indeed affected due to lack of propionylation rather than other lysine modifications,

we increased H2BK17pr by treating S2 cells with propionate. We assessed 6 genes (*Cks3OA*, *mr*, *SkpB*, *Sdic2*, *lds* and *SCAP*) that show decreased mRNA level in H2BK17A flies and 2 genes (*pie* and *Traf_like*) that show increased mRNA level (Fig. 5m). All of these genes are bound by H2BK17pr, implicating that they are direct targets of this modification (Supplementary Data 4). For the former group, propionate treatment increases their expression either significantly or with a very

**Fig. 4 | H2BK17pr regulates proteostasis. a** Histograms show the total protein level of WT and H2BK17A fly heads under AL (upper panel) or TRF condition (lower panel) at the indicated ZT (*n* = 4 biologically independent experiments, two-tailed Mann–Whitney *U* test for unpaired comparisons, * *p* = 0.02857). **b** Histograms show the weight of WT and H2BK17A flies under AL (upper panel) or TRF condition (lower panel) at the indicated ZT (*n* = 4 biologically independent experiments, two-tailed Mann–Whitney *U* test for unpaired comparisons, * *p* = 0.02857). **c** Histogram shows the glucose level of WT and H2BK17A flies (*n* = 4 biologically independent experiments, two-tailed Mann–Whitney *U* test for unpaired comparisons). **d** Histogram shows the triglyceride level of WT and H2BK17A flies (*n* = 5 biologically independent experiments, two-tailed Mann–Whitney *U* test for unpaired comparisons, ** *p* = 0.007937). **e** Histogram shows the yeast consumption of WT and H2BK17A flies measured by CAFÉ assay (*n* = 4 biologically independent experiments, two-tailed Mann–Whitney *U* test for unpaired comparisons, * *p* = 0.02857). **f** Histogram shows the sucrose consumption of WT and H2BK17A flies measured by CAFÉ assay (*n* = 5 biologically independent experiments, two-tailed Mann–Whitney *U* test for unpaired comparisons). **g** Histograms show total protein level of S2 cells treated with sodium propionate and sodium acetate or equal volume of PBS as control

(*n* = 4 biologically independent experiments, two-tailed Mann–Whitney *U* test for unpaired comparisons, * *p* = 0.02857, ** *p* = 0.006895, *** *p* = 0.000008). **h** Western blots of protein extracts from S2 cells treated with sodium propionate and sodium acetate at indicated concentrations for 24 h. ACTB is used as a loading control. **i** Histogram demonstrates total protein level of AC16 cells treated with sodium acetate or sodium propionate at 5 mM for 48 h, or equal volume of PBS as control (*n* = 4 biologically independent experiments, two-tailed Mann–Whitney *U* test for unpaired comparisons, * *p* = 0.02857). **j** Histogram demonstrates relative protein level of U2OS and HEK293 cells treated with sodium acetate and sodium propionate for 48 h at indicated concentrations, or equal volume of PBS as control (*n* = 3 biologically independent experiments, two-tailed Mann–Whitney *U* test for unpaired comparisons). The protein level of PBS treatment is set to 1. **k** Western blots of protein extracts from AC16 cells treated with sodium propionate or sodium acetate under indicated concentrations. ACTB is used as a loading control. Data are presented as the mean ± SD. AC sodium acetate, PR sodium propionate. The blotting experiments were conducted with three independent repeats with similar results (**h** and **k**). Source data are provided as a Source Data file.

---

prominent trend for all of the genes except *Cks3OA*. For the latter group, propionate significantly reduces the expression of *Traf_like* and increases the expression of *pie* although this does not quite reach statistical significance. In summary, we believe that the majority of the UPS DEGs (and perhaps other DEGs as well) are indeed influenced by H2BK17pr.

## H2BK17pr mediates the rhythmic regulation of TRF on proteasomal genes

The effect of TRF on H2BK17pr oscillation suggest a role for H2BK17pr in mediating the influence of TRF on rhythmic processes (Supplementary Fig. 6a–e). Since TRF is known to promote daily rhythm of gene expression, we tested whether H2BK17pr is involved in modulating rhythmic gene expression in response to TRF[42]. To address this, we conducted circadian analysis on our RNA-seq data and computationally identified 1628, 1265, 1489 and 932 cycling genes from TRF-WT, AL-WT, TRF-H2BK17A and AL-H2BK17A flies, respectively (Fig. 6a; Supplementary Data 3c–f). TRF induces rhythmic expression of 1129 genes in WT, while H2BK17A results in loss of rhythmicity for 69% of the genes under TRF condition. GO-based enrichment analyses revealed that genes rhythmically regulated by TRF and no longer rhythmic in H2BK17A are involved in proteasome-mediated ubiquitin-dependent protein catabolic (PUPC) process and circadian rhythm (pathways present in "TRF-WT & not AL-WT" and "TRF-WT & not TRF-H2BK17A", but not in "TRF-H2BK17A & not AL-H2BK17A", Fig. 6b). These findings suggest that TRF drives rhythmic expression of genes in these two pathways, and this process is mediated by H2BK17pr.

Next, we performed circadian analysis of all ChIP-seq data, and found that H2BK17pr rhythmically binds to only 80 genes under AL and 1368 genes under TRF (Fig. 6c; Supplementary Data 4c and 4d). This implicates that TRF not only promotes cyclic propionylation of the H2BK17 site, but also dramatically enhances rhythmic binding of propionylated H2B in the genome. We then conducted GO-based enrichment analysis on the 1368 genes that are rhythmically bound by H2BK17pr under TRF (Fig. 6d). We noticed that the phosphorylation pathway is enriched, which is well known to play crucial roles in circadian regulation[34,43–45]. The 1368 genes rhythmically bound by H2BK17pr under TRF condition were hierarchically categorized into 3 distinct clusters based on their temporal pattern (Supplementary Fig. 10d). For each cluster, motif enrichment analysis was performed to identify TFs that potentially participate in regulating genes rhythmically bound by H2BK17pr (Supplementary Fig. 10; Supplementary Data 4e). Taken together, our ChIP-seq analyses indicate that TRF strongly enhances the binding capacity of H2BK17pr to modulate rhythmic transcription.

To identify genes directly regulated by H2BK17pr in a cyclic manner under TRF, we focused on genes that meet these two criteria: (1) rhythmically bound by H2BK17pr under TRF, identified from ChIP-seq data; (2) the expression is rhythmic under TRF but not AL, derived from RNA-seq data. We anticipated these genes should be directly regulated by the cyclic activities of H2BK17pr, mediating the effects of TRF on the oscillation of the transcriptome. There are 118 genes that meet the two criteria, most of which exhibit peak expression between Zeitgeber Time 6-12 (ZT6-12, ZT0 is the time of lights on), consistent with the peak of H2BK17pr level (Fig. 6e, f and Supplementary Fig. 6b; Supplementary Data 3g). In particular, we found H2BK17pr binding with robust rhythm at the locus of *discs overgrown* (*dco*, also known as *doubletime*), which encodes a protein kinase that phosphorylates PER and is critical for timing the clock (Fig. 6g)[46]. Next, we compared the temporal binding profile of H2BK17pr with the expression profile of the 118 genes and found that 97 display significant correlation, 86 of which are significant positive correlation, implying that H2BK17pr acts to enhance the transcription of these genes (Supplementary Data 3h). Again, we found these genes were enriched in pathways including circadian rhythm and PUPC process (Fig. 6h). In the circadian rhythm pathway, most of genes enriched are proteasomal genes which are also enriched in the PUPC pathway, including *proteasome β6 subunit* (*Prosbeta6*), *proteasome β5 subunit* (*Prosbeta5*), *regulatory particle non-ATPase 12* (*Rpn12*), *suppressor of exocyst mutations 1* (*Sem1*) and *proteasome β2 subunit* (*Prosbeta2*) (Supplementary Data 3g).

Given that all the pathway enrichment analyses pointed to a role for H2BK17pr in directly modulating the cyclic expression of genes involved in the PUPC process, we next focused on these genes which include *Prosbeta6*, *Prosbeta5*, *Prosbeta2*, *Rnp12* and *Sem1*. H2BK17A abolishes the mRNA oscillation of *Prosbeta6*, *Prosbeta2*, *Rnp12* and *Sem1* (Fig. 6i). For *Prosbeta5*, although there is still significant rhythm of mRNA profile in H2BK17A, the amplitude tends to be dampened compared to WT (14.9 in WT vs. 9.9 in H2BK17A, Supplementary Data 3c and 3e). To verify these genes are indeed affected due to lack of propionylation, we treated S2 cells with propionate and found that this significantly increases the mRNA level of these genes, implicating that H2BK17pr promotes the expression of these genes (Fig. 6j).

Overall, this series of results indicate that TRF induces rhythmic expression of genes involved in the PUPC process which in turn influences protein metabolism, and it is highly likely that H2BK17pr mediates the effects of TRF on these proteasomal genes.

## H2BK17pr displays distinctive temporal distribution profile in the genome

To further delineate the genomic mechanism by which H2BK17pr influences transcription, we compared the occupancy of H2BK17pr

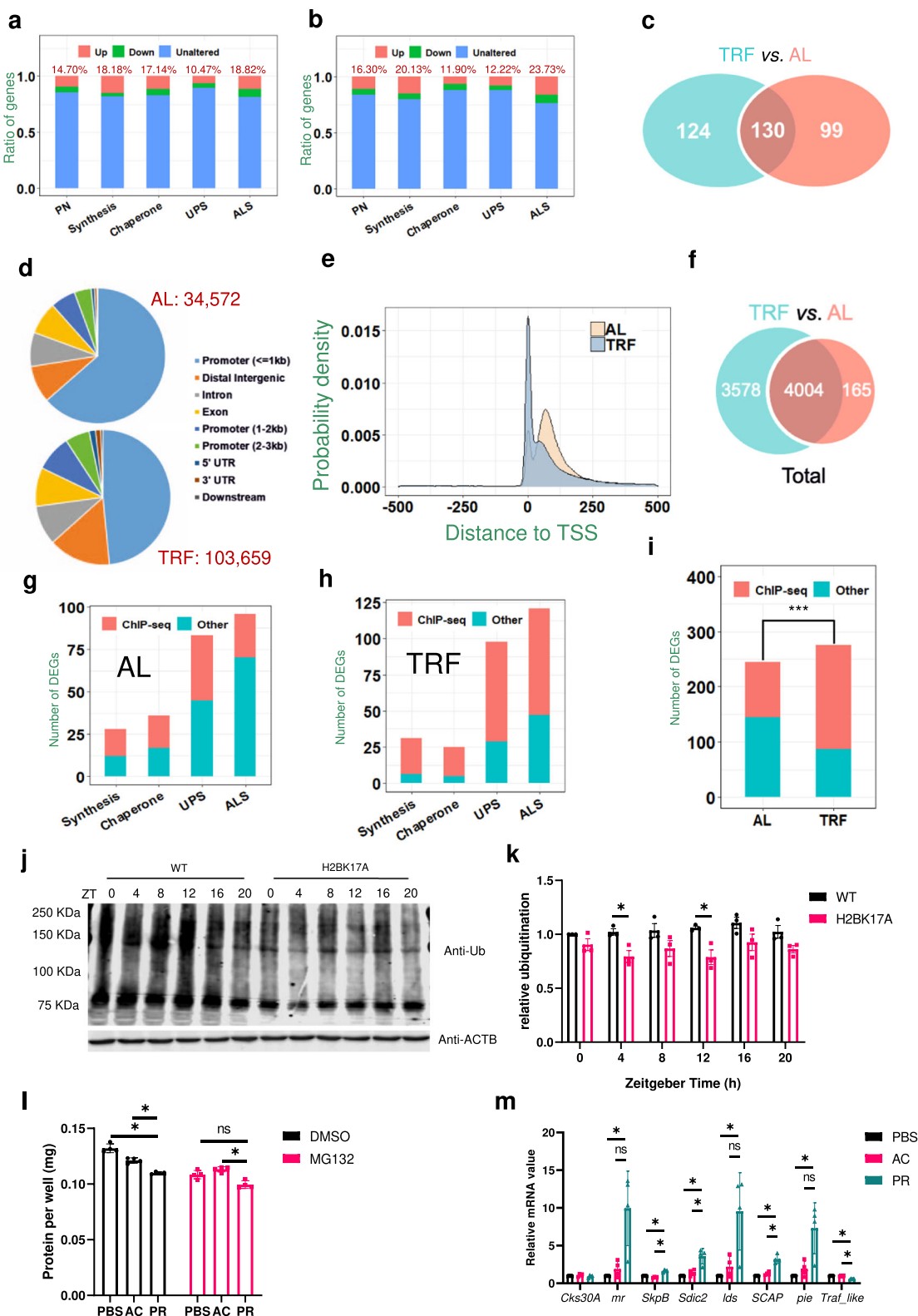

with that of H2B and H3K27ac, a well-known marker of active transcription[47]. ChIP-seq analysis was performed for TRF-treated fly heads using H2B or H3K27ac antibody. $1.05 \times 10^8$ and $9.64 \times 10^7$ clean reads were obtained from H2B- and H3K27ac-enriched fly heads with average numbers of 135 and 3863 mappable genes, respectively. H2B binding peaks that are located near TSS could only be identified at 195 genes, 75 of which are also bound by H2BK17pr (Supplementary

Fig. 13a). Only 3 of the 195 genes show rhythmic H2B binding. In addition, Spearman's correlations were calculated for 36 genes that display both H2B and H2BK17pr peaks at 3 or more time points. One gene exhibits significant positive correlation while 5 genes show significant negative correlation (Supplementary Fig. 13b).

5136 genes display H3K27ac binding peaks near TSS, among which 4372 genes are also occupied by H2BK17pr (Supplementary Fig. 13c).

**Fig. 5 | H2BK17pr determines global protein level by regulating UPS genes.**
**a**, **b** Ratios of PN DEGs under AL (**a**) and TRF (**b**) conditions. **c** Number of PN DEGs under TRF vs. AL. **d** The number of H2BK17pr binding peaks in the genome and genomic distribution of the peaks. UTR, untranslated region. **e** The distribution of peaks surrounding transcription start site (TSS) in AL- and TRF-treated samples. **f** Number of genes bound by H2BK17pr in AL- vs. TRF-treated samples. **g**, **h** Number of PN DEGs bound by H2BK17pr under AL (**g**) and TRF (**h**). Red represents DEGs identified by ChIP-seq to be bound by H2BK17pr, while green represents the other DEGs not bound by H2BK17pr. **i** Number of PN DEGs bound by H2BK17pr under AL vs. TRF. Red represents DEGs identified by ChIP-seq to be bound by H2BK17pr, while green represents the other DEGs not bound by H2BK17pr. Chi-Squared Test, *** $p = 3.74E{-}10$. **j** Representative Western blots of proteins from whole-head extracts of WT and H2BK17A flies and collected at the indicated ZT during LD. ACTB is used as a loading control. The blotting experiments were conducted with three independent repeats with similar results. **k** Histogram shows quantification of relative ubiquitin level of blots in (**j**). The average intensity of WT at ZT0 is set to 1 ($n = 3$ biologically independent experiments, two-tailed Student's $t$ test, * $p = 0.02151$ and $0.01770$ from left to right). **l** Histogram shows total protein level of S2 cells treated with DMSO or proteasome inhibitor MG132, along with 5 mM sodium propionate, 5 mM sodium acetate or PBS for 24 h ($n = 4$ biologically independent experiments, two-tailed Mann–Whitney $U$ test for unpaired comparisons, * $p = 0.02857$). **m** Histogram shows relative mRNA level of indicated genes in S2 cells treated with sodium acetate, sodium propionate for 24 h at 5 mM or equal volume of PBS as control ($n = 4$ biologically independent experiments, two-tailed Mann–Whitney $U$ test for unpaired comparisons, from left to right, * $p = 0.02107$). The mRNA level of PBS treatment is set to 1. Data are presented as the mean ± SD. AC, sodium acetate. PR, sodium propionate. Source data are provided as a Source Data file.

137 genes exhibit cyclic H3K27ac binding, which is ~2.7% of genes with H3K27ac peaks, much lower than the 18% for H2BK17pr (1368 out of 7562 genes display rhythmic H2BK17pr binding). We further analyzed the temporal expression profile of the 137 genes rhythmically bound by H3K27ac and found that only 15 genes display cycling at the mRNA level, much less than that of H2BK17pr (118 genes are rhythmically expressed and bound by H2BK17pr in a cyclic manner). There is only one overlap (CG5168) with the 118 genes potentially targeted by H2BK17pr (Supplementary Fig. 13d and 13e; Supplementary Data 3h). The proteasomal genes that we are particularly interested in display rhythmic binding of H2BK17pr but not H3K27ac (Supplementary Fig. 13e). In addition, correlation analysis was conducted for 3384 genes that show both H3K27ac and H2BK17pr peaks at 3 or more time points. Only 71 (2.10%) of the genes exhibit positive correlations and 102 (3.01%) display negative correlations (Supplementary Fig. 13f).

In summary, these results reveal co-occupancy of H2BK17pr and H3K27ac at the majority of the genes with these modifications. On the other hand, the temporal pattern of H2BK17pr distribution near TSS shares little similarity with that of H2B or H3K27ac, indicating that the oscillation of H2BK17pr occupancy is not triggered by rhythmic binding of H2B or H3K27ac.

### The potential regulatory mechanism of H2BK17pr

Based on published studies, protein propionylation is believed to be controlled by lysine acetyltransferases (KATs) and lysine deacetylases (KDACs)[48]. In flies, there are 5 known KATs and 10 known KDACs (including 5 histone deacetylases and 5 sirtuins, Supplementary Data 6). Thus we screened these enzymes for ones involved in regulating propionylation of H2BK17. We treated S2 cells with pharmacological inhibitors targeting these enzymes and examined the effects on H2BK17pr level (Figs 7a, b and Supplementary Fig. 14a). We found that Entinostat and BG45, inhibitors of type I histone deacetylase (HDAC1 and 3 in flies), increase H2BK17pr in a dose-dependent manner, suggesting a role for HDAC1 and/or HDAC3 in reducing the propionylation at H2BK17. To test whether HDAC1 or HDAC3 regulates H2BK17pr, we knocked down *HDAC1* or *HDAC3* in S2 cells but observed no significant effect on H2BK17pr level (Supplementary Fig. 14b–d). On the other hand, A485, inhibitor of histone acetyltransferases (HATs) p300/CBP decreases H2BK17pr in a dose-dependent manner, implicating that p300/CBP acts to increase H2BK17 propionylation (Fig. 7a and b). We further validated this by knocking down *nejire* (*nej*), the fly ortholog of mammalian *CBP* and *p300*, and observed significant reduction of H2BK17pr (Supplementary Fig. 14e–g). In addition, we found that inhibitor of histone acetyltransferases TIP60/MOF (MG149) decreases H2BK17pr in a dose-dependent manner, suggesting that these two enzymes may also facilitate the propionylation of H2BK17 (Fig. 7a and b). We knocked down *Tip60* and *mof* but failed to observe significant alteration of H2BK17pr level (Supplementary Fig. 14h–k). Furthermore, p300/CBP inhibitor A485 modestly but significantly elevates total protein level, whereas HDAC1/3 inhibitor Entinostat

substantially decreases total protein level, consistent with the effects of these drugs on H2BK17pr level (Supplementary Fig. 14l). Taken together, these results support a role of KATs and KDACs in regulating H2BK17pr[17].

For a better understanding of how PN is regulated by H2BK17pr, we integrated our own multi-omics data and experimental findings in this study along with the knowledge from public data resources to computationally model a H2BK17pr-centered signaling network (Fig. 7c; Supplementary Data 7). This network contains the 3 KATs (NEJ, TIP60 and MOF) and 2 KDACs (HDAC1 and HDAC3) that potentially regulate H2BK17pr, the 172 H2BK17pr-bound PN DEGs, and 14 UPS genes that may be directly regulated by H2BK17pr in a cyclic manner under TRF which include 2 genes that also belong to the 172 PN DEGs (Figs. 5, 6, 7a and 7b). In addition, we included 10 TFs that may regulate these PN genes and 8 TFs that may regulate the cycling genes identified by motif enrichment analysis and transcription regulation prediction (Supplementary Fig. 10e; Supplementary Data 7). In total, we obtained 688 transcriptional regulations for 184 genes (Supplementary Data 7). Interestingly, we found that among the 18 TFs predicted is the core clock protein CLOCK (CLK), and one of the genes involved in proteolysis, CG4538, is predicted to be regulated by CLK (Fig. 7c; Supplementary Data 7).

## Discussion

Here in this study we present the largest dataset to our knowledge of Kpr in eukaryotes. We found that propionylated proteins are enriched in the mitochondria and nucleosomes in flies, consistent with acylation observed in other organisms and further confirming the evolutionary conservedness of these modifications[13, 17]. Temporally regulated Kpr occurs mostly on proteins involved in mitochondrial metabolic processes, which is similar to Kac and implies a major role for propionylation in modulating the rhythmicity of mitochondrial metabolism[24].

KprFunc predicted H2BK17pr to be functionally important, and we indeed identified a role for this modification in regulating proteostasis. This role of H2BK17pr may be evolutionarily conserved, as the equivalent site on human H2B (H2BK23) is also propionylated. Moreover, propionate treatment reduces total protein level in AC16 cells, which is likely accompanied by elevated H2BK23pr level. Although all four groups of PN genes were altered in H2BK17A flies, only inhibition of the proteasome blocked the reduction of total protein level induced by propionate treatment, indicating that propionate treatment reduces protein level by enhancing proteasome-mediated degradation. This is consistent with the observation of decreased global ubiquitination in H2BK17A flies. Piecing these results together, we propose that H2BK17pr promotes protein degradation by enhancing the expression of UPS genes. Accelerated oxidation of branched-chain amino acids (BCAAs; leucine, isoleucine and valine) have long been known to be associated with protein catabolic states, such as during sepsis and trauma, but the mechanism appears to be undercharacterized[49,50]. Propionyl-CoA is an intermediate of isoleucine and

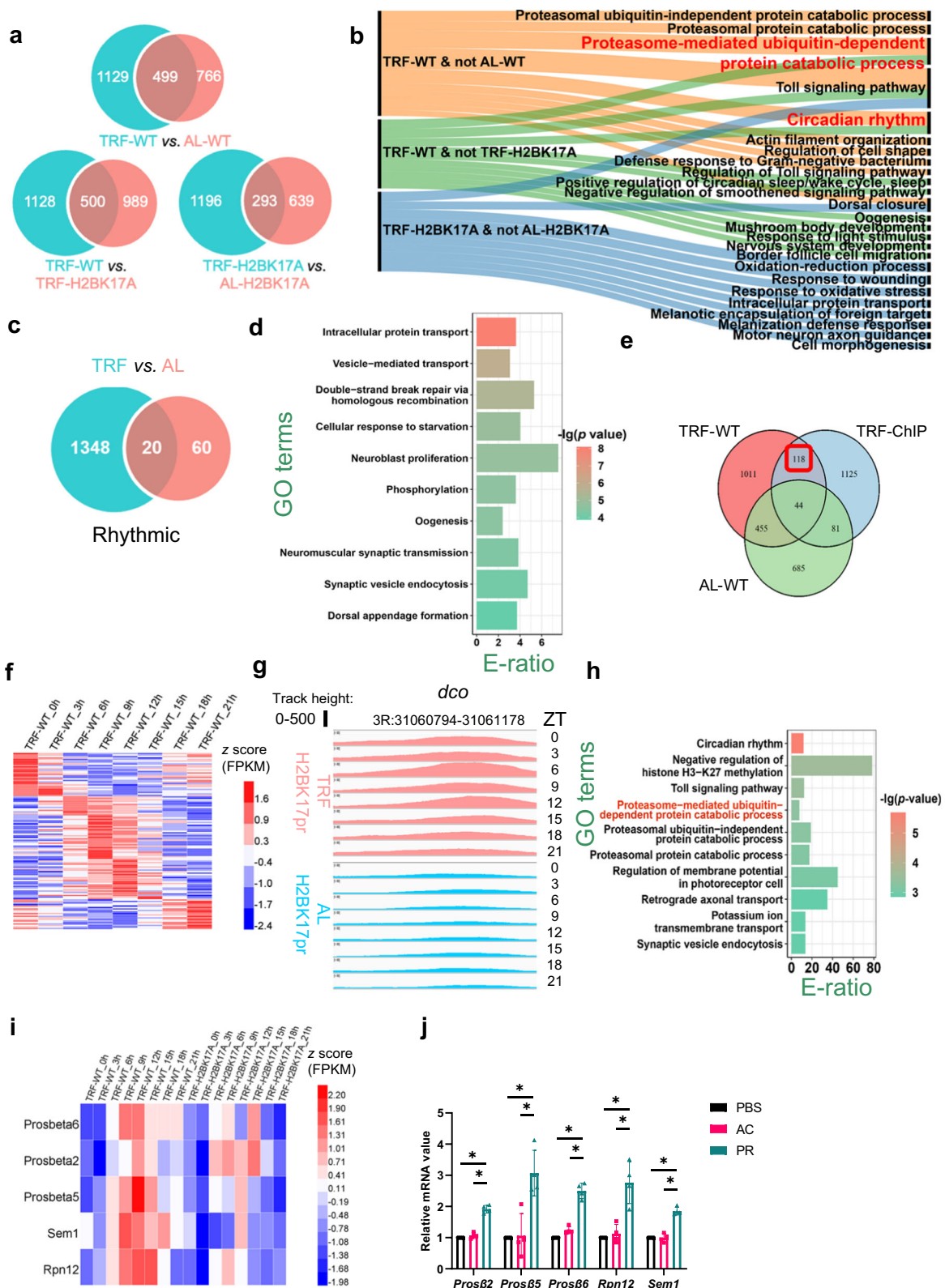

valine oxidation, and here we identify a role for propionylation in enhancing protein degradation. Therefore, we propose that propionyl-CoA/propionylation serves as a catabolic signal, driving the system into a protein catabolic state. Consistent with this idea, the excessive protein catabolism in PA and MMA may be triggered by accumulation of propionyl-CoA and enhanced propionylation, possibly via a mechanism dependent on H2B propionylation. BCAA catabolism has

also been linked to maple syrup urine disease, isovaleric acidemia, liver cirrhosis, chronic renal failure and muscle wasting, and we suspect altered propionylation may contribute to the development of these diseases/conditions as well[49]. On the other hand, lack of H2BK17pr as in the case of H2BK17A also results in altered protein metabolism. To better understand what increase in total protein level means for the H2BK17A flies, we assessed their protein intake. We found these flies

**Fig. 6 | H2BK17pr mediates the rhythmic regulation of TRF on proteasomal genes. a** The number of cycling genes identified from RNA-seq samples of WT and H2BK17A flies during AL or TRF. **b** GO-based enrichment analysis of cycling genes identified in TRF-WT samples but not AL-WT samples, TRF-WT samples but not TRF-H2BK17A samples and TRF-H2BK17A samples but not AL-H2BK17A samples, respectively. E-ratio, enrichment ratio (one-sided hypergeometric test). **c** Number of genes rhythmically bound by H2BK17pr in AL- vs. TRF-treated samples. **d** GO-based enrichment analysis of 1386 genes rhythmically bound by H2BK17pr (one-sided hypergeometric test). **e** 118 genes were found to be rhythmically bound by H2BK17pr under TRF, and cycling under TRF but not AL. **f** The normalized FPKM values of the 118 genes at indicated ZT under TRF. **g** Integrative Genomics Viewer

(IGV) snapshots depicting H2BK17pr binding at *dco* promoter region of WT fly heads under TRF and AL condition. **h** GO-based enrichment analysis of the 118 genes (one-sided hypergeometric test). **i** The normalized FPKM values of the 5 proteasomal genes in PUPC pathway at the indicated ZT. **j** Histogram shows relative mRNA level of indicated genes in S2 cells treated with sodium acetate (AC) or sodium propionate (PR) for 24 h at 5 mM or equal volume of PBS as control ($n = 4$ biologically independent experiments, two-tailed Mann–Whitney $U$ test for unpaired comparisons. PBS vs. PR, *$p = 0.02107$; AC vs. PR, *$p = 0.02857$). The mRNA level of PBS treatment is set to 1. Data are presented as the mean ± SD. Source data are provided as a Source Data file.

show specific reduction of yeast intake (which is the major source of protein in fly food) but not sugar, as well as a decrease of metabolites in protein digestion/absorption. We believe this is a consequence of reduced protein catabolism and elevated protein level, i.e., the flies eat less protein in response to the perturbed proteostasis. It is noteworthy that besides propionylation, acetylation, butyrylation, crotonylation, hydroxyl-butyrylation and di-methylation have also been reported to occur at H2BK23, while the fly H2BK17 site can be acetylated[17,28,51,52]. Currently, we have not quite ruled out the influences of these other modifications on proteostasis, and an interesting future direction would be to investigate the function of these modifications at H2BK17/H2BK23 and their interactions with propionylation.

To understand the dynamic modulation of H2BK17pr in a physiological context, we further explored the regulatory mechanism and function of its daily cycling. Our results indicate that feeding cycles rhythmically regulate H2BK17pr, similar to other PTMs that are tightly linked with metabolic state such as Kac and O-GlcNAcylation[23,38]. While rhythmic feeding is known to be driven by the circadian clock, here we found that circadian clock appears to exert direct influences on H2BK17pr that is independent of feeding[38]. In WT flies, restricting feeding to the natural feeding time (during the day) enhances the cycling of H2BK17pr whereas feeding during unnatural feeding time (in the night) abolishes this cycling. This suggests that the circadian clock buffers the influence of nighttime feeding to prevent excessive H2BK17pr at non-optimal time of the day. What's more, the rhythmic modulation of TRF on H2BK17pr requires the clock, as in *per0* flies H2BK17pr does not show significant oscillation even under daytime restricted feeding condition. These observations are strikingly similar with the cyclic pattern of O-GlcNAcylation in flies, in which case both the clock and TRF regulate rhythmic expression and activities of enzymes responsible for catalyzing O-GlcNAcylation and generating the donor for this modification[38]. Taken together, we believe that feeding cycles provides a major drive for rhythmic propionylation at H2BK17, while circadian clock gates this process so the influences of feeding are prominent only during certain periods of the day.

The strengthening of H2BK17pr oscillation by daytime restricted feeding suggests a role for this rhythmic modification in mediating the impact of TRF. Stronger evidence supporting this hypothesis is that under daytime restricted feeding, the number of rhythmic H2BK17pr binding sites increased almost 17 times (Fig. 6c). Based on our analysis, H2BK17pr rhythmically binds to 118 genes that are cyclically regulated by TRF. In addition, for 97 out of the 118 genes, significant correlation between the temporal expression pattern of the gene and temporal binding pattern of H2BK17pr at its genomic loci was observed (Supplementary Data 3h). Most (86/97) of the correlations are positive, implying a role for H2BK17pr in facilitating gene expression which is consistent with the previous study that reported histone propionylation stimulates transcription in vitro[53]. This is further supported by the observation that most of the genes bound by H2BK17pr are also occupied by the active transcription marker H3K27ac (Supplementary Fig. 13c).

The rhythmic influences of H2BK17pr on proteasomal genes implicate a role for H2BK17pr in modulating the daily variation of

proteasomal activity. Diurnal rhythm of proteasomal activity has been reported in mouse liver, which is speculated to be driven by the availability of ubiquitinated substrates that also show diurnal oscillation[54]. Our results suggest that rhythmic expression of proteasomal components may be another mechanism underlying the cycling of proteasomal activity. One caveat is that although H2BK17pr and proteasomal genes show daily cycling, we did not observe temporal variation of total protein level in WT or H2BK17A flies. We believe this actually exemplifies how circadian rhythm modulates proteostasis: rendering the proteasomal activity to be higher at certain times of the day when there are more proteins that need to be degraded, and vice versa. Therefore, although there is a diurnal rhythm in proteasomal flux, the total protein level remains relatively stable throughout the day.

All three major families of HATs (GNATs, MYSTs and p300/CBP) have been shown to function as propionyltransferases, while sirtuins SIRT1-3 function as depropionylases[17,18]. Our pharmacological screen uncovered p300/CBP/NEJ, TIP60/MOF and class I HDAC (HDAC1/3) to be potentially regulating propionylation level at H2BK17. RNA interference experiments further validated the role of NEJ as a propionyltransferase, but knocking down *HDAC1* or *HDAC3* did not alter H2BK17pr level. This could be due to functional redundancy of HDACs in S2 cells, or that the extent of knock down is not sufficient to disturb enzymatic activity at H2BK17. Given that inhibitors of HDAC1 or HDAC3 do not alter H2BK17pr level, we favor the former rather than the latter explanation. Knocking down either *Tip60* or *mof* does not alter H2BK17pr level, and further characterization will be needed to confirm the role of TIP60/MOF in H2BK17 propionylation. Notably, a very recent study also identified p300 and class I HDAC (HDAC1 and HDAC2) as regulatory enzymes of histone β-hydroxybutyrylation in human cell culture[55]. Although in this study, we have not conducted in vitro assays to demonstrate that NEJ and HDAC1/3 directly catalyze propionylation at the H2BK17 site, given what is known regarding the roles of these enzymes in regulating acylation, we believe it is likely that they propionylate/depropionylate H2BK17.

Based on our previous proteomic data and the RNA-seq results in this study, MOF exhibits significant oscillation at the protein level, but none of the other 2 KATs or KDACs identified displays significant oscillation at mRNA, protein or phosphorylation level[34]. We suspect these KATs and KDACs are rhythmically recruited to the genome and propionylate/depropionylate H2BK17, as they may interact with TFs that are regulated in a cyclic manner. Among the TFs predicted, CLK has been shown to rhythmically bind to at least 800 genomic sites[56]. The mRNA levels of *Rfx* display significant oscillation under TRF, while Motif 1 Binding Protein (M1BP) cycles at the protein level (Supplementary Data 3c)[34]. It is also possible that the clock and TRF modulate the activities of the KATs and KDACs at the post-translational level, similar with the enzymes involved in O-GlcNAcylation[38]. Together, these different layers of regulations potentially contribute to the rhythm of H2BK17pr.

In conclusion, here we uncovered a dynamic Kpr landscape and using a small-sample learning framework KprFunc, we identified H2BK17pr which we further validated to be evolutionarily conserved

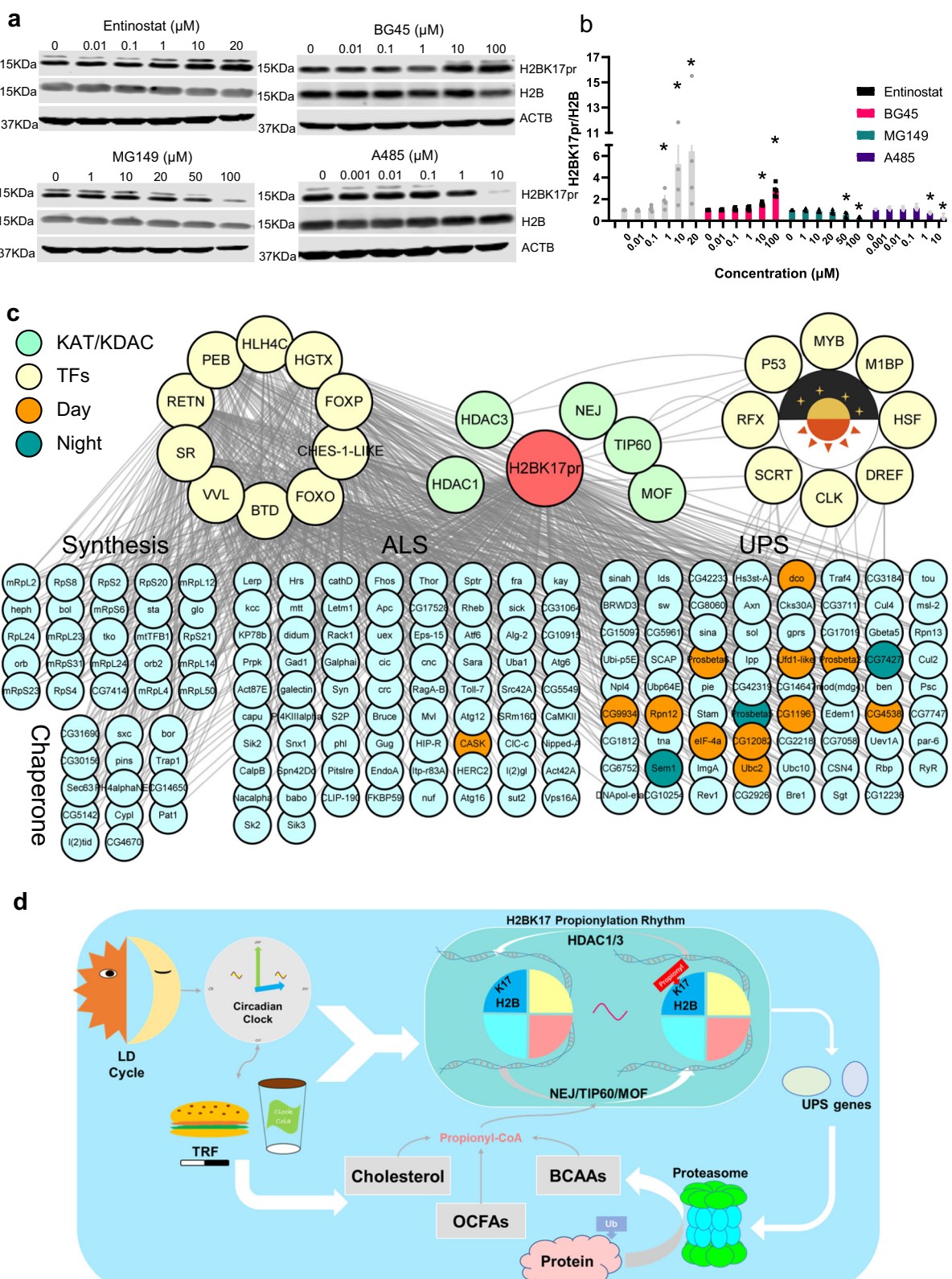

and functionally important. Based on our results, we propose a model regarding the regulatory mechanism and physiological function of H2BK17pr: feeding cycle drive the oscillation of H2BK17pr in a circadian clock-dependent manner, which in turn regulates the expression of UPS genes, contributing to daily oscillation of protein catabolic activities (Fig. 7d). Restricting feeding to the day time further promotes rhythmic H2BK17pr and thus rhythmic expression of

proteasomal genes, helping to ensure that protein catabolic processes occur at the optimal time of the day.

## Methods

### Flies and cell lines

The fly strains and genotypes used in this manuscript are listed (Supplementary Data 8). All flies were reared on standard

**Fig. 7 | The potential regulatory mechanism of H2BK17pr. a** Representative Western blots of protein extracts from S2 cells treated with Entinostat, BG45, MG149 and A485 under indicated concentrations. ACTB is used as a loading control. The blotting experiments were conducted with three independent repeats. **b** Histogram shows quantification of normalized H2BK17pr level in (**a**). The normalized intensity at 0 μM is set to 1 (two-tailed Mann–Whitney $U$ test for unpaired comparisons. Entinostat, $n = 4$ biologically independent experiments, * $p = 0.02107$; BG45, $n = 5$ biologically independent experiments, * $p = 0.00749$; MG149, $n = 4$ biologically independent experiments, * $p = 0.02107$; A485, $n = 4$ biologically independent experiments, * $p = 0.02107$). **c** A H2BK17pr-centered signaling network including the 5 potential regulators (in light green) and PN genes differentially expressed in H2BK17A flies, as well as 18 TFs (in yellow) predicted to be involved. The 10 TFs on the left were predicted to regulate genes with constant H2BK17pr binding while the 8 TFs on the right were predicted to regulate genes with rhythmic H2BK17pr binding. Orange circle indicates genes with peak H2BK17pr binding during the day, while dark green circle indicates genes with peak H2BK17pr binding at night. **d** Model illustrating the regulatory mechanism of H2BK17pr, which in turn modulates the expression of UPS genes that ultimately influence protein degradation. Data are presented as the mean ± SD. Source data are provided as a Source Data file.

cornmeal–yeast–sucrose medium and kept in light/dark (LD) cycles at 25 °C. Food were accessible *ad libitum*. For propionylomic study, male $w^{1118}$ (Bloomington Stock Center, BL3605) flies were crossed with $y^1, w^*$ and $y^1, w^*, per^0$.[25] For RNA-seq, ChIP-seq and metabolomics studies, male $w^{1118}$ and H2BK17A flies were used. For the rest of the study, H2BK17A was backcrossed onto the isogenic $w^{1118}$ background (Bloomington Stock Center, #5905) for five generations and male flies were used along with isogenic $w^{1118}$ as control.

*Drosophila* S2 cells used in this study is a kind gift from Dr. Xi Zhou (Wuhan University). Human AC16 cells is a kind gift from Dr. Chengqi Xu (HUST). NIH 3T3, U2OS and HEK293 cells used in this study were from ATCC.

### Fly husbandry and head collection
Flies were collected within 3–7 days of eclosion and entrained in LD for 3 days. After that, they are transferred into constant darkness (DD). Flies were collected during LD or on the first day of DD and frozen at −80 °C. Frozen flies were vortexed for 10 s to separate the head from the body.

For TRF, 1–7-day-old flies were entrained under LD for 2 days, then split into TRF group and AL group. TRF groups were fed from Zeitgeber Time 0 (ZT0, time of lights on) to ZT12 of from ZT12 to ZT24, respectively. They are kept on 1% agar during fasting period. TRF and AL groups were fed under these conditions for more than 1 week and then collected for analysis.

### Propionylomic profiling
Flies were collected within 3–7 days of eclosion and entrained in LD schedule for 3 days. After that, flies were transferred into DD. On the first day of DD, flies were collected every 6 h and frozen at −80 °C. Frozen flies were vortexed for 10 s to separate the head from the body. Fly heads were homogenized by sonication for 5 min in urea lysis buffer [8 M urea, 1× proteinase inhibitor and phosphatase inhibitor (4693116001 and 04906845001, Roche), 2 mM EDTA]. Homogenates were centrifuged at $20,000 \times g$ for 10 min and supernatants were collected. High-abundance proteins were removed with ProteoMiner Protein Enrichment Kit (1633007, Bio-Rad). Finally, protein concentration was measured by BCA Protein Assay Kit (23227, Thermo Scientific) and adjusted to be consistent across the time series.

To digest the proteins, protein solution was first treated with 5 mM dithiothreitol (DTT) at 56 °C for 30 min, followed by alkylation with 11 mM iodoacetamide for 15 min at room temperature in the dark. After that, 100 mM triethylammonium bicarbonate (TEAB) was used to dilute the protein samples to reduce the concentration of urea to <2 M. Two trypsin digestions were performed, using the mass ratio of 1:50 trypsin-to-protein for overnight treatment and 1:100 for 4 h, respectively. After digestion, Strata X C18 SPE column (8B-S001-HBJ, Phenomenex) and vacuum-dry were used to desalt the peptides and then the peptides were reconstituted with 0.5 M TEAB. The peptides were subsequently processed by TMT kit. The peptides were incubated with labeling reagent at room temperature for 2 h, followed by desalting and vacuum drying. The peptides were then fractionated by high pH reverse-phase HPLC using Agilent 300Extend C18 column (5 μm particles, 4.6 mm ID, 250 mm length). Briefly, peptides were first separated with a gradient of 8–32% acetonitrile (pH 9.0) over 60 min into 60 fractions. Then, the 60 fractions were combined into 8 fractions at 8 intervals and dried by vacuum centrifuging (Supplementary Data 1e).

To enrich propionylated peptides, peptides were dissolved in NETN buffer (100 mM NaCl, 1 mM EDTA, 50 mM Tris-HCl, 0.5% NP-40, pH 8.0) and incubated with pre-washed antibody-conjugated beads (SKU: PTM-202, PTM Bio) at 4 °C overnight with gentle shaking. Then the beads were washed four times with NETN buffer and twice with $H_2O$. The bound peptides were eluted from the beads with 0.1% trifluoroacetic acid. Finally, the eluted fractions were combined and vacuum-dried. For LC-MS/MS analysis, the resulting peptides were desalted with C18 ZipTips (ZTC18S096, Millipore) according to the manufacturer's instructions.

The peptides were dissolved in solvent A (0.1% formic acid and 2% acetonitrile) and directly loaded onto a home-made reversed-phase analytical column. Peptides were separated with a gradient from 7% to 20% solvent B (0.1% formic acid in 90% acetonitrile) over 24 min, 20–35% in 8 min and climbing to 80% in 5 min then holding at 80% for the last 3 min, all at a constant flow rate of 300 nL/min on an EASY-nLC 1000 UPLC system (Thermo Fisher Scientific). The separated peptides were analyzed in Q ExactiveTM Plus (Thermo Fisher Scientific) with a nano-electrospray ion source. The electrospray voltage applied was 2.0 kV. The full MS scan resolution was set to 70,000 for a scan range of 350–1800 $m/z$. Up to 20 most abundant precursors were then selected for further MS/MS analyses with 15 s dynamic exclusion. The HCD fragmentation was performed at a normalized collision energy (NCE) of 30%. The fragments were detected in the Orbitrap at a resolution of 17,500. Fixed first mass was set as 100 $m/z$. Automatic gain control (AGC) target was set at 5E4, with an intensity threshold of 5E3 and a maximum injection time of 200 ms. The number of technical and/or biological replicates is 1 for propionylomic profiling ($n = 1$).

### Database search for propionylated peptides
MaxQuant (v.1.5.3.30)[26] was used for standard database search of MS/MS raw data. MS/MS spectra were searched against *D. melanogaster* proteome database obtained from UniProt (Version 201706)[57], which contained 13,558 unique fly protein sequences. For each protein, the decoy sequence was automatically generated and combined with the reference database. The digestion mode was set to Specific and Trypsin/P was chosen as cleavage enzyme allowing up to two missed cleavages. Carbamidomethyl (C) was the fixed modification, while oxidation (M), acetyl (protein N-term) and propionyl defined by Unimod (http://www.unimod.org/), a database of protein modifications[58], were the variable modifications. Seven was set as the minimum peptide length and 4600 Da as the maximum peptide mass. The false discovery rates for PSM, protein and Kpr site decoy fraction were all set to <1% and minimum score for modified peptides was set to >40. To further evaluate the specificity of the enrichment of Kpr-containing peptides, database search was re-performed by further including acetyl (K) as an additional variable modification. Then, the MS/MS spectra of all modified peptides and un-modified peptides were visualized and annotated in MaxQuant[26]. The diagnostic ion at $m/z$ 140.1075 for Kpr was manually checked for each propionylated peptide.

## Acetyl-CoA and propionyl-CoA quantification by ELISA

Flies were collected within 3–7 days of eclosion and maintained on daytime TRF condition for at least 7 days. Then, flies were collected every 3 h and frozen at −80 °C. Frozen flies were vortexed for 10 s to separate the head from the body. The acetyl-CoA and propionyl-CoA level of WT fly heads were measured by using acetyl-CoA (JM-11679M2, JINGMEI Biotech.) and propionyl-CoA ELISA kit (JM-12736M2, JINGMEI Biotech.), following the manufacturer's instructions. Plate absorbance was measured with spectrophotometer Multiskan GO (Thermo Fisher Scientific) at 450 nm.

## Hypergeometric test

For GO-based enrichment analysis of propionylated proteins, GO annotation files were downloaded on 18 November 2020 from the Gene Ontology Resource (http://geneontology.org/)[59] and contained 12,616 fly proteins with at least one GO term. For each GO term $t$, we defined the following:

$N$ = number of proteins annotated by at least one GO term.

$n$ = number of proteins annotated by GO term $t$.

$M$ = number of propionylated proteins annotated by at least one GO term.

$m$ = number of propionylated proteins annotated by GO term $t$.

The enrichment ratio (E-ratio) of $t$ was then computed, and the $p$ was calculated with the hypergeometric distribution as follows:

$$\mathrm{E-ratio} = \frac{m}{M} \bigg/ \frac{n}{N} \qquad (1)$$

$$p = \sum_{m'=m}^{n} \frac{\binom{M}{m'}\binom{N-M}{n-m'}}{\binom{N}{n}} (\mathrm{E-ratio} \geq 1), \text{ or} \qquad (2)$$

$$p = \sum_{m'=0}^{m} \frac{\binom{M}{m'}\binom{N-M}{n-m'}}{\binom{N}{n}} (\mathrm{E-ratio} < 1) \qquad (3)$$

The hypergeometric test was also adopted for GO-based enrichment of cycling genes identified from ChIP-seq and RNA-seq, and Kyoto Encyclopedia of Genes and Genomes (KEGG)-based enrichment analyses of metabolites. The annotation file (released on 7 March 2021) containing 6182 metabolites involved in at least one fly pathway was downloaded from the ftp server of KEGG (ftp://ftp.bioinfromatics.jp/)[60].

## Preparation of benchmark data sets for training KprFunc

Besides the 344 Kpr sites quantified in this study, we further collected 1129 known and non-redundant Kpr sites in 606 eukaryotic or prokaryotic proteins, from a previously developed database, Compendium of Protein Lysine Modifications (CPLM 4.0)[27]. Recently, Ding et al. released a propionylomic profiling of 292 Kpr sites in 142 proteins from *Danio rerio*[31]. The three datasets were integrated, and redundancy was removed. Then we defined a propionylation site peptide PSP($m$, $n$) as a lysine residue flanked by $m$ upstream residues and $n$ downstream residues. As previously described[32], the PSP(30,30) items around known Kpr sites were regarded as positive data, while PSP(30,30) items from other non-Kpr lysine residues were used as negative data. For multiple identical PSP(30,30) items in positive or negative data, only one item was reserved. After redundancy clearance, our positive dataset contained 1707 Kpr sites in 890 proteins (Supplementary Data 2b).

In addition, we manually re-checked Kpr sites collected from CPLM 4.0[27] and found 12 Kpr sites with known important functions. SOD2K132pr in *D. rerio* was recently reported to be functionally important and thus was also included[31]. Altogether, these 13 Kpr sites were used as the secondary positive data for KprFunc to predict the functional relevance of Kpr sites (Supplementary Data 2a), while other known Kpr sites were used as the secondary negative data.

## The KprFunc algorithm

The implementation of KprFunc comprised three steps, including sequence embedding, deep learning-based pre-training of the initial model, KprFunc-I, for prediction of Kpr sites, and MAML for fine-tuning the final model, KprFunc, for prediction of the functional relevance of Kpr sites.

(i) Sequence embedding. In GPS 5.0[32], PWD determines the weight of each position in the phosphorylated peptide, and SMO optimizes the scoring matrix from an initial amino acid substitution matrix (e.g., BLOSUM62) with a pairwise identity of no more than 62%[61]. The optimized weights and matrix were then used by the final model to calculate average similarity scores between peptides, which results in higher performance in distinguishing phosphorylated vs. non-phosphorylated sites compared to the original model. In this study, such a strategy was not directly used for prediction, but adopted to embed PSP(30, 30) items into 1D vectors, which were then iteratively optimized by PLR, a classical machine learning algorithm, prior to model training.

First, the average similarity score ($S$) between a given PSP(30, 30) item $P$ and all PSP(30, 30) items in positive data was defined as:

$$S = \frac{1}{N} \sum_{j=-30}^{L-31} \left( \sum_{i=1}^{N} M[P_j, K_{ij}] \right) \times W_j \qquad (4)$$

$L$ denotes the length of the PSP(30, 30) item and is equal to 61 in this study. $N$ is the number of known Kpr sites in the positive data set. $K_{ij}$ is the amino acid at position $j$ around a known Kpr site $K_i$ ($i$ = 1, 2, 3, …, $N$). $W_j$ represents the weight value of position $j$, and initially equal to 1. $M$ is an amino acid substitution matrix, and BLOUSM62 was selected as the initial matrix.

Next, PWD was used to optimize $W_j$ values, and SMO was used to optimize $M$, with the help of PLR with the ridge (L2) penalty. The two procedures were iteratively repeated, and the inverse of regularization strength (C) was increased by 0.1 each time to avoid over-fitting. For each iteration, the 10-fold cross-validation was conducted to evaluate the performance. Receiver operating characteristic (ROC) curves were drawn, and AUC values were determined. The iteration was terminated until the AUC value was no longer increased. Using optimized $W_j$ values and $M$, each PSP(30,30) item $P$ was embedded into a 1D vector as below:

$$\mathbf{V_P} = (S_{AA}, S_{AC}, S_{AD}, \cdots, S_{**})_{300} \qquad (5)$$

$S_{ab}$ was calculated as below:

$$S_{ab} = \frac{1}{N} \sum_{j=-30}^{L-31} C_j \times M_{optimized}[a,b] \times W_j \qquad (6)$$

$C_j$ is the number of an amino acid pair ($a$, $b$) at position $j$, and $W_j$ is the optimized weight for position $j$. In final scoring matrix $M_{optimized}$, there were 24 types of characters including 20 types of amino acids and 4 non-canonical characters (B, Z, X, and *). In each vector, there were (24 + 1)*24/2 = 300 variables as informative features. To avoid over-fitting during feature representation, the ridge (L2) penalty of PLR was adopted, using the "sag" solver in the class LogisticRegressionCV of scikit-learn v0.21.0 (https://scikit-learn.org/), a widely used machine-learning toolbox[62].

(ii) Deep learning-based pre-training. We implemented a 4-layer DNN framework, and each layer contained a number of computational units named neurons. To avoid over-fitting during model training, we used the Dropout method, which randomly discarded nodes from the hidden layers if the performance was enhanced. In each layer, all reserved neurons were used for feature representation and output. For model training, the first layer acted as the input layer to receive all the embedded $V_P$ vectors. The following two hidden layers were fully

connected to extract and represent features. Multiple activation functions were tested, while the simplest linear unit (LU) activation function was adopted and defined as below:

$$LU(x) = x \qquad (7)$$

The last layer was taken as the output layer, which contained a sigmoid neuron to output a score for a given $\mathbf{V_P}$ vector. The sigmoid function is defined as below:

$$Sigmoid(y) = \frac{1}{1 + e^{-y}} \qquad (8)$$

The sigmoid score ranged from 0 to 1, and a higher sigmoid score represents a higher probability of a PSP(30,30) item to be a real Kpr site. The 4-layer DNN model was implemented in Keras 2.4.3 (http://github.com/fchollet/keras), on a computer with the NVIDIA GeForce RTX 3090 GPU, a Genuine Intel (R) CPU @ 2.30 GHz CPU, and 128 GB RAM. When training the model of KprFunc-i, transient parameters such as the learning rate, epoch, mini-batch size, and dropout probability were iteratively optimized (Supplementary Data 2c).

(iii) MAML for fine-tuning. MAML is a state-of-the-art AI technology, and especially useful for solving new learning tasks using training data with a very small size[33]. In contrast to ab initio training a model, here the initial model, KprFunc-i, was adopted for fine-tuning with the 13 known functional Kpr sites. During the fine-tuning stage, we randomly sampled secondary negative data with a ratio of 5:1 to the secondary positive data, and 5 models were then separately constructed using such a data set through the fivefold cross-validations. Such a process was repeatedly performed to gradually optimize the model, and terminated if the AUC value no longer increased.

## Performance evaluation and comparison

For the evaluation of KprFunc, the true positive (TP), true negative (TN), false positive (FP) and false negative (FN) values were calculated for each model. Then, four widely used measurements, including the sensitivity (Sn), Sp, accuracy (Ac) and Mathew correlation coefficient (MCC) were calculated as below:

$$Sn = \frac{TP}{TP + FN} \qquad (9)$$

$$Sp = \frac{TN}{TN + FP} \qquad (10)$$

$$Ac = \frac{TP + TN}{TP + FP + TN + FN} \qquad (11)$$

$$MCC = \frac{(TP \times TN) - (FN \times FP)}{\sqrt{(TP + FN) \times (TN + FP) \times (TP + FP) \times (TN + FN)}} \qquad (12)$$

The 10- and 5-fold cross-validations were performed to evaluate the performance of KprFunc-i and KprFunc, respectively. The ROC curve was illustrated based on the final Sn and 1-Sp scores, and the AUC value was computed. For each predictor, three thresholds including high, medium and low were selected based on Sp values of 95%, 90% and 85%, respectively (Supplementary Data 2d). Confusion matrices were then generated under the medium threshold.

Additional prediction models were construed using other methods, including GPS 5.0[32], and 4 machine learning methods, DNN, SVMs, RF and KNN, without sequence embedding. Moreover, DNN, SVMs, RF, and KNN algorithms were used for model training by directly using the

13 known functional Kpr sites without pre-training. The prediction models of SVMs, RF, and KNN were also implemented in scikit-learn v0.21.0[62].

## Implementation of the online service

The online service of KprFunc was developed with PHP and JavaScript at http://kprfunc.biocuckoo.cn/, in an easy-to-use manner. For prediction of general or functional Kpr sites, three thresholds including "High", "Medium" and "Low" could be selected based on Sp values of 95%, 90% and 85%, respectively. We extensively tested the online service on various web browsers including Microsoft Edge, Google Chrome, and Mozilla Firefox. More details are available at http://kprfunc.biocuckoo.cn/userguide.php.

## Prediction of functional relevant Kpr sites using KprFunc

Prior to further experimental validation, we used KprFunc to score all the 344 Kpr sites quantified in this study (Supplementary Data 1a), and the high threshold was selected (Supplementary Data 2d). Previously, we conducted time-course proteomic profiling of fly heads, and systematically identified 620 high-quality circadian proteins[34]. In this study, KprFunc was used to predict potential Kpr sites with functional relevance in these proteins. Again, the high threshold was adopted.

## Structural analysis of H2BK17pr

The predicted 3D structure of fly H2B protein was downloaded from AlphaFold Protein Structure Database (https://www.alphafold.ebi.ac.uk/)[63] as a PDB file. An open-source software package, PyMol 2.5.0 (https://pymol.org/2/), was used for visualization of fly H2B and calculation of distances between H2BK17pr and other residues. Only residues within ≤4 Å were reserved for visualization.

## Evolutionary analysis of H2B proteins and H2BK17pr

The 7,671 H2B protein sequences in HistoneDB 2.0 (https://www.ncbi.nlm.nih.gov/research/HistoneDB2.0/index.fcgi/browse/)[36] were downloaded as a FASTA file. To reconstruct a rooted phylogenetic tree, the H3.3 protein in *Saccharomyces cerevisiae* was taken as the outgroup sequence. The multi-alignment of all sequences was performed by MEGA 7.0.26[64], using the MUSCLE method, and a neighbor-joining tree was constructed. For simplicity, several typical H2Bs were shown in Fig. 3b, as well as the yeast H3.3. The bootstrap method was selected as the test of phylogeny for 10,000 times and the Poisson model was adopted for substitution. Lastly, the conservation of H2BK17 site was directly judged from the multi-alignment result. For convenience, the multi-alignment result of 38 *Drosophila* H2B proteins was shown in Supplementary Fig. 4a. Conserved residues of K17, K23 and K23 in H2Bs from *D. melanogaster*, *Homo sapiens* and *Mus musculus* were illustrated using DOG 2.0[65], along with functional domains in these proteins.

## Generation of H2BK17pr antibody

Two peptides, "CAGKAQ-(propionyl)K-NITKTDK" (peptide 1, from Ala12 to Lys24 of H2B) and "KAGKAQ-(propionyl)K-NITKTC" (peptide 2, from Lys11 to Thr22 of H2B) were synthesized and immunized in New Zealand rabbit (PTM BIO). Animal studies were approved by the Committee on Animal Research and Ethics of PTM BIO.

## Generation of H2BK17A fly line

H2BK17A fly line was generated according to previously published methods[66]. In brief, H2BK17A mutation was introduced into WT histone gene repeat units (His-GU) construct. pUAST-attB plasmid containing 5 copies of mutant H2B was then generated. *His^D* male flies were crossed with PhiC31 virgin flies. 0–1 h embryos were harvested for microinjection. About 1000 embryos were injected. PCR was used to verify double attP/attB integrations.

### H2BK17A mutation sequencing validation

Mutant fly was homogenized with DNA sequencing extract buffer [10 mM Tris-HCl at pH 8.0, 1 mM EDTA at pH 8.0, 25 mM NaCl and 0.02% proteinase K (CW2584, CWBIO)] and incubated at 37 °C for 30 min. PCR was performed with Taq Plus MasterMix (CW2849, CWbiotech). The PCR reaction was performed as following: 94 °C for 2 min followed by 94 °C for 10 s, 57 °C for 15 s and 72 °C for 1 min 20 s for 35 cycles. The primers used were listed (Supplementary Data 8). PCR products were sequenced by using commercial Sanger sequencing (AuGCT Co.).

### Protein extraction and Western blot

Proteins were extracted from fly heads and cultured cells using SDS lysis buffer [50 mM Tris-HCl at pH 7.6, 150 mM NaCl, 1% Triton X-100, 0.5% SDS, 1 mM EDTA at pH8.0, 2 mM DTT, 330 nM TSA, 10 mM Nicotinamide, 1 × proteinase inhibitor and phosphatase inhibitor (4693116001 and 04906845001, Roche)]. After homogenization, protein lysates were centrifuged at $12,000 \times g$ for 15 min at 4 °C and incubated at 95 °C in loading buffer for 5 min. Equal amounts of protein were loaded into each well on 12% SDS-PAGE gels and then transferred to nitrocellulose membranes for 2 h at 90 V. Membranes were incubated with primary antibody at 4 °C overnight, followed by secondary antibody at room temperature for 1 h.

The primary antibodies used were as follows: pan anti-propionylation (1:1000, PTM-201, PTM BIO, CN), anti-H2BK17pr, anti-H2B (1:1000, ab52484, Abcam, US), anti-H2BK23ac (1:1000, PTM-174, PTM BIO, CN), anti-H3K27ac (1:1000, ab4729, Abcam, US), anti-ACTB (1:5000, AC026, ABclonal, CN), Anti-ubiquitination (1:2000,sc-8017, SCBT, US) and Anti-ATG8A (1:1000, ab109364, Abcam, US). Donkey secondary antibodies (1:10000 dilution) were conjugated either with IRDye 680 or IRDye 800 (926-68072 and 926-32213, LI-COR Biosciences, US) and visualized with an Odyssey Infrared Imaging System (LI-COR Biosciences). For quantification of pan ubiquitination, intensity of the whole lane was quantified. Uncropped blots are provided in the Source Data file.

### Cell culture and transfection

S2 cells were cultured in Schneider's *Drosophila* Medium containing 10% FBS and 1% Penicillin-Streptomycin. NIH 3T3, U2OS, AC16 and HEK293 cells were cultured in DMEM containing 10% FBS and Penicillin-Streptomycin. S2 cells were plated in 12-well plates and transfected with riboFECT™CP Transfection Kit (C10511, RiboBio) following manufacturer's instructions. Cells were harvested 24–48 h after transfection. Mammalian cells and *Drosophila* cells were plated to 70–90% confluent, sodium propionate and sodium acetate were mixed with PBS and adjusted to equal final volume, and then added into the medium to culture for 24–48 h. 100 μM Cycloheximide and 40 μM MG132 were added into the medium of S2 cells 8 h before harvesting, 95 μM chloroquine was added into the medium of S2 cells for 48 h before harvesting. The siRNA sequences used are listed in Supplementary Data 8.

### Immunoprecipitation followed by MS (IP-MS) analysis

For immunoprecipitation of fly heads, 150 mg male fly heads were homogenized with 1 ml PBS buffer and then centrifuged at $5000 \times g$ for 10 min at 4 °C. Pellets were suspended with 500 μl IP buffer [50 mM Tris-HCl at pH 7.4, 150 mM NaCl, 1%NP-40, 0.1% SDS, 2 mM EDTA at pH8.0, 0.15 mM spermine, 0.5 mM spermidine, 330 nM TSA, 10 mM Nicotinamide, 1× proteinase inhibitor and phosphatase inhibitor (4693116001 and 04906845001, Roche)]. For immunoprecipitation of cultured cells, $1 \times 10^7$ cells were rinsed with 1 ml PBS buffer followed by homogenization with 500 μl IP buffer. Homogenates were then sonicated for $18 \times 5$ s at 10% power and centrifuged at $17,000 \times g$ for 5 min at 4 °C. 20 μl of 450 μl supernatant was used as input, and the remaining was divided into two parts. 5 μg H2BK17pr antibody or

rabbit normal IgG were added to the two parts, respectively. The samples were incubated at 4 °C overnight and then 50 μl of Protein A magnetic beads (1614013, Bio-Rad) were added, followed by 4 h incubation at 4 °C. Beads were washed with IP buffer for $3 \times 10$ min, and protein was eluted by SDS–PAGE loading buffer for Western blotting.

For in-gel tryptic digestion, gel pieces were destained in 50 mM NH$_4$HCO$_3$ in 50% acetonitrile ($v/v$) until clear. Gel pieces were dehydrated with 100 μl 100% acetonitrile for 5 min. The liquid was then removed, and the gel pieces were rehydrated in 10 mM DTT and incubated at 56 °C for 60 min. Gel pieces were again dehydrated in 100% acetonitrile. Liquid was removed and gel pieces were rehydrated with 55 mM iodoacetamide. Samples were incubated at room temperature in the dark for 45 min. Gel pieces were washed with 50 mM NH$_4$HCO$_3$ and dehydrated with 100% acetonitrile. Gel pieces were rehydrated with 10 ng/μl trypsin resuspended in 50 mM NH$_4$HCO$_3$ and incubated on ice for 1 h. Excess liquid was removed and gel pieces were digested with 200 μl of 20 ng/μl trypsin at 37 °C overnight. Peptides were extracted with 50% acetonitrile/5% formic acid, followed by 100% acetonitrile. Peptides were dried completely and resuspended in 2% acetonitrile/0.1% formic acid.

The peptides were dissolved in 0.1% formic acid (solvent A) and directly loaded onto a home-made reversed-phase analytical column (15 cm length, 75 μm i.d.) packed with 1.9 μm/120 Å ReproSil-PurC18 resins (Dr. Maisch GmbH, Ammerbuch, Germany). The gradient was comprised of an increase from 6% to 23% solvent B (0.1% formic acid in 98% acetonitrile) over 16 min, 23% to 35% in 8 min and climbing to 80% in 3 min then holding at 80% for the last 3 min, all at a constant flow rate of 400 nl/min on an EASY-nLC 1000 UPLC system (Thermo Fisher Scientific).

The peptides were subjected to NSI source followed by tandem mass spectrometry (MS/MS) in Q Exactive™ Plus (Thermo) coupled online to the UPLC. The electrospray voltage applied was 2.0 kV. The $m/z$ scan range was 350–1800 for full scan, and intact peptides were detected in the Orbitrap at a resolution of 70,000. Peptides were then selected for MS/MS using NCE setting at 28 and the fragments were detected in the Orbitrap at a resolution of 17,500. One MS scan was followed by 20 MS/MS scans with 15 s dynamic exclusion. Automatic gain control (AGC) was set at 5E4. The number of technical and/or biological replicates was 1 for IP-MS analysis ($n = 1$).

### qRT-PCR

RNA was extracted from −80 °C frozen fly heads and cells. 200 fly heads per sample or $1 \times 10^7$ cells per sample were homogenized in Total RNA Isolation (TRIzol) Reagent (15596026, Invitrogen), by using a handheld motor with plastic beads. After mixing with trichloromethane, homogenates were centrifuged at $12,000 \times g$ and suspension was precipitated with 75% ethanol. After air dry, total RNA was treated with RQ1 DNase (M6101, Promega) to remove genomic DNA and stored at −80 °C.

Quantitative real-time PCR was performed with One-Step RT-PCR SuperMix (AT141, Transgen). The PCR reaction was performed as follows: 45 °C for 5 min; 94 °C for 2 min; 94 °C for 5 s, 60 °C for 15 s, 72 °C for 20 s for 40 cycles (Applied Biosystems). The ΔΔCT method was used for quantification. *Beta-Actin* was used as internal control. The primers used are listed in Supplementary Data 8.

### Locomotor activity monitoring and analysis

Locomotor activity levels of adult male flies were monitored by *Drosophila* Activity Monitoring System (TriKinetics) for 7 days of LD followed by 7 days of DD. For DD rhythmicity, chi-squared periodogram analyses were performed by Clocklab (Actimetrics). Rhythmic flies were defined as those in which the chi-squared power was ≥10 above the significance line. Period calculations considered all flies with rhythmic power ≥10. Dead flies were defined by 0 activity on DD7 and removed from analysis.

## Assessment of metabolic indices

Weight, glucose, triglycerides and protein level of 1-week old male adult flies were detected following a published protocol[67]. In brief, 25 flies were weighed by electronic weighing balance with a resolution of 0.1 mg. 25 flies were frozen and vortexed for 10 s to separate the head from the body. Heads were homogenized with 160 μl and 1 ml 1% Triton X-100, respectively. The homogenates were centrifuged at 5000 × $g$ for 10 min at 4 °C and supernatants were measured with Bradford Assay Kit (23227, Thermo Fisher Scientific) following manufacturer's instructions. 1 μl hemolymph extracted from 40 flies was mixed with 19 μl TBS (137 mM NaCl, 2.7 mM KCl, 5 mM Tris pH 6.6) and measured with D-Glucose Content Assay Kit (AKSU001C, Boxbio) following manufacturer's instructions. Triglycerides of 25 flies were extracted and measured with Triglyceride Content Assay kit (AKFA003M, Boxbio) following manufacturer's instructions.

## Drosophila CApillary FEeder (CAFÉ) assay

Food consumption of male flies was measured by CAFÉ assay following previously published methods[68]. In brief, 25 flies were entrained under LD cycle at 25 °C for 3 days and transferred to the CAFÉ system at ZT0. 5% yeast extract was added into capillary tube, with 1% agar at the bottom of the vials as water supplement. Food consumption was measured every hour from ZT4 to ZT 12. Vials without flies were used as evaporation control.

## RNA-seq profiling and data analysis

200 fly heads per sample were homogenized in Total RNA Isolation (TRIzol) Reagent (15596026, Invitrogen), by using a handheld motor with plastic pestle. After mixing with trichloromethane, homogenates were centrifuged at 12,000 × $g$ for 15 min at 4 °C and suspension was precipitated with 75% ethanol. After air dry, total RNA was treated with RQ1 DNase (M6101, Promega) to remove genomic DNA and stored at −80 °C.

Before sequencing, RNA purity was measured by NanoPhotometer (IMPLEN, CA, USA), whereas the concentration and integrity of RNA were assessed by Qubit RNA Assay Kit in Qubit 2.0 Flurometer (Life Technologies) and RNA Nano 6000 Assay Kit of the Agilent Bioanalyzer 2100 system (Agilent Technologies). Agarose gel electrophoresis was also conducted for all samples. 1.5 μg RNA per sample was used to construct the libraries for mRNA sequencing with NEBNext Ultra RNA Library Prep Kit for Illumina (E7770, NEB). Poly-T oligo-attached magnetic beads were employed to purify mRNA from total RNA, and then fragmentation was carried out with divalent cations under elevated temperature in NEBNext First Strand Synthesis Reaction Buffer (5×). First and second strand complementary DNAs (cDNAs) were then synthesized by using random hexamer primers. For hybridization, NEBNext Adaptor with hairpin loop structure was ligated onto the 3′ ends of the fragments, which were then purified by AMPure XP system (Beckman Coulter) to control for the length. PCR was performed after treating the cDNA with USER Enzyme for 15 min at 37 °C followed by 95 °C for 5 min. The purity of PCR products and assessment of library quality were performed on the Agilent Bioanalyzer 2100 system. In order to perform cluster generation of the index-coded samples, HiSeq 4000 PE Cluster Kit (PE-410-1001, Illumia) was used on a cBot Cluster Generation System. Sequencing of library preparations was performed on Illumina NovaSeq 6000 platform.

For the analysis of RNA-seq data, raw reads were first mapped to the reference genome of *D. melanogaster*, which was downloaded from Ensembl (release version 90, http://www.ensembl.rog/)[69]. Then the software packages of Bowtie2 (version 2.2.4) and TopHat (version 2.1.1) were used to generate the BAM files. Cufflinks (version 2.2.1) was employed to assemble the reads and calculate the expression levels of individual mRNAs based on FPKM values[70]. Based on the results of Cufflinks, genes with significant changes of FPKM values between AL-WT vs. AL-H2BK17A samples or TRF-WT vs. TRF-H2BK17A samples were identified as DEGs ($p < 0.01$). R package pheatmap was adopted for the two-way hierarchical clustering and heatmap visualization of RNA-seq samples.

## Integration of genes in the *Drosophila* PN

Four types of PN genes, including protein synthesis factors, molecular chaperones, UPS enzymes/regulators, and ALS regulators, were collected in *D. melanogaster*. First, fly proteins annotated with "translation" (GO:0006412), "protein anabolism" (GO:0043037), "protein biosynthesis" (GO:0006416), "protein biosynthetic process" (GO:0006453) and/or "regulation of translation" (GO:0006417) were collected as protein synthesis factors. From a previous study[7], we obtained 332 human molecular chaperones, and their potential orthologs were computationally identified by reciprocal best hits (RBHs)[71] in *D. melanogaster*, using the blastall program from the BLAST software package[72]. Moreover, mouse UPS and ALS regulators identified by Gallotta et al.[73] were downloaded, and their corresponding fly orthologs were identified by RBHs. Previously, we developed two comprehensive databases, iUUCD[74] and THANATOS[75]. Fly UPS enzymes/regulators were directly taken from the former, whereas fly ALS regulators were obtained from the latter. In addition, fly proteins with GO annotations containing the keyword of "proteasome" or "proteasomal" were also integrated as UPS regulators. Finally, all these genes were merged together, and redundant genes were removed to prepare the benchmark data set for the *Drosophila* PN (Supplementary Data 3b).

## ChIP-seq profiling and data analysis

1500 fly heads per sample were homogenized with ice-cold PBS plus buffer [1× PBS buffer supplemented with 330 nM Trichostatin A (TSA), 10 mM Nicotinamide, 1× proteinase inhibitor and phosphatase inhibitor (4693116001 and 04906845001, Roche)]. The homogenates were centrifuged at 5000 × $g$ for 10 min at 4 °C to remove cell debris. Pellets containing the nuclei were resuspended in 500 ml PBS buffer with 1% formaldehyde and crosslinked for 15 min at room temperature, then stopped by adding 150 μl 1 M glycine and rotation for 10 min. Pellets collected by centrifugation at 5000 × $g$ for 10 min at 4 °C were washed with 1 ml PBS plus buffer. Pellets containing the nuclei were resuspended in NEB buffer (10 mM Tris-HCl at pH 8.0, 10 mM NaCl, 0.5 mM EDTA at pH8.0, 0.1 mM EGTA at pH8.0, 1 mM DTT, 0.5% NP-10, 0.5 mM spermidine, 0.15 mM spermine, 330 nM TSA, 10 mM Nicotinamide, 1× proteinase inhibitor and phosphatase inhibitor (Roche) followed by centrifugation at 17,000 × $g$ for 10 min at 4 °C. Pellets were resuspended in 300 μl sonication buffer (10 mM Tris-HCl at pH 8.0, 2 mM EDTA at pH8.0, 0.6% SDS, 0.2% Triton X-100, 0.5 mM spermidine, 0.15 mM spermine, 330 nM TSA, 10 mM Nicotinamide, 1× proteinase inhibitor and phosphatase inhibitor) to release the nuclei. Samples were sonicated with 650 W Φ3 sonicator (JY92-IIN, Scientz) for 120 × 5 s on 35% power and then centrifuged at 17,000 × $g$ for 5 min at 4 °C, supernatants were recentrifuged at 17,000 × $g$ for 10 min at 4 °C. 240 μl of the supernatants were mixed with 720 μl IP buffer (10 mM Tris-HCl at pH 8.0, 150 mM NaCl, 2 mM EDTA at pH8.0, 0.5 mM EGTA at pH 8.0, 1% Triton X-100, 0.1% sodium deoxycholate, 1 × proteinase inhibitor and phosphatase inhibitor). Mixed supernatants were divided into two parts for IP with specific antibody and IgG, respectively. H2BK17pr, H2B (ab52484, Abcam) and H3K27ac (ab4729, Abcam) antibodies were used for ChIP-seq. 5 μg antibody or normal IgG from same species [mouse normal IgG (sc-2025, SCBT, US), rabbit normal IgG (2729, CST, US)] were then added and incubated overnight at 4 °C. 50 μl of Protein A magnetic bead slurry (1614013, Bio-Rad) were washed twice with 1 ml of NEB buffer and added into IP and IgG samples and incubated for 4 h with rotation at 4 °C. Next, beads were captured and washed in 1 ml low-salt RIPA buffer (50 mM Tris-HCl at pH 7.4, 150 mM NaCl, 1% Triton X-100, 0.1% SDS, 0.1% sodium deoxycholate and 1 mM EDTA at pH 8.0), 1 ml high-salt RIPA buffer (50 mM

Tris-HCl at pH 7.4, 500 mM NaCl, 1% Triton X-100, 0.1% SDS, 0.1% sodium deoxycholate and 1 mM EDTA at pH 8.0), 1 ml LW buffer (10 mM Tris−HCl at pH 8.0, 0.25 M LiCl, 0.5% NP-40, 0.5% sodium deoxycholate, 1 mM EDTA) and 1 ml TE buffer (10 mM Tris-HCl at pH 8.0 and 1 mM EDTA at pH 8.0) for 10 min sequentially. After that, 150 μl of ChIP Elution buffer [50 mM Tris−HCl at pH 8.0, 10 mM EDTA, 1% SDS, 1 mM DTT, 0.1 mg/ml proteinase K (CW2584, CWBIO), 50 mM NaCl, and 0.05 mg/ml RNase A (CW0600, CWBIO)] were added and incubated for 2 h at 37 °C. Beads were then removed from specific antibody and IgG samples, and supernatants were de-crosslinked overnight at 65 °C. DNA was extracted using QIAquick PCR Purification Kit (28104, Qiagen).

ChIP DNA degradation and contamination was monitored on agarose gels. DNA purity was assessed by the NanoPhotometer® spectrophotometer (IMPLEN). DNA concentration was measured by Qubit® DNA Assay Kit (Q32854, Life Technologies) in Qubit® 3.0 Flurometer (Life Technologies). The purified DNA was used to construct ChIP-seq library and pair-end sequencing was performed on Illumina platform. Library quality was assessed on the Agilent Bioanalyzer 2100 system.

For analysis of the ChIP-seq data, Bowtie2 (version 2.2.4)[70] was adopted for reads mapping to the reference genome of *D. melanogaster* downloaded from Ensembl[69]. Then, Samtools (version 1.3.1)[76] was used to sort the BAM files and generate the SAM files. The peak calling was performed by MACS2 (version 2.1.4) with the $q$ value threshold of $10^{-5}$[77], and peak annotation was conducted with ChIPseeker[78]. H2BK17pr binding signals were searched from sequences −2000 to 500 bp surrounding TSSs of genes to annotate peaks at promoters[79]. Lastly, identified peaks from different time points were merged by the program mergePeaks in HOMER (version 4.11)[80], while the maximum distance was set to 120. Before further analysis, pileup of each peak was normalized using total reads. R package pheatmap was adopted for the two-way hierarchical clustering, and diagram of heatmaps for ChIP-seq samples, as well as genes with rhythmic bindings under TRF-treatment. For each cluster, the program findMotifsGenome in HOMER (version 4.11)[80] was used for the motif enrichment analysis, and top 100 enriched motifs were acquired. For each enriched motif, the *Drosophila* ortholog or homolog of the corresponding TF in the same family was determined by searching a comprehensive database, AnimalTFDB 3.0[81].

## Metabolomic profiling and data analysis

In order to increase extraction efficiency, we used methanol/water and acetate/methanol to extract metabolites sequentially. 50 mg of fly heads per sample were homogenized with 500 μl ice-cold methanol/water (70%, $v/v$). Homogenates were vortexed for 5 min and incubated on ice for 15 min, followed by centrifugation at 12,000 *rpm* at 4 °C for 10 min. 400 μl supernatant was collected into tube 1. 500 μl ethyl acetate/methanol (1:3, $v/v$) was added into the original centrifuge tube, and the mixture was oscillated for 5 min and incubated on ice for 15 min. The mixture was centrifuged at 12,000 *rpm* at 4 °C for 10 min and 400 μl of supernatant was collected into tube 2. Tube 1 and 2 were combined and the 800 μl supernatants were vacuum dried. Then 100 μl 70% methanol was added into the dried product and sonicated for 3 min. Finally, the samples were centrifuged at 12,000 *rpm* for 3 min at 4 °C, and 60 μl supernatant was used for LC-MS/MS analysis.

All samples were analyzed by the LC-MS/MS system. The analytical conditions were as follows. UPLC: column, Waters ACQUITY UPLC HSS T3 C18 (1.8 μm, 2.1 mm*100 mm); column temperature: 35 °C; flow rate: 0.3 ml/min; injection volume: 1 μl; solvent system: water (0.01% methanolic acid): acetonitrile; gradient program of positive ion ($v/v$): 95:5 at 0 min, 79:21 at 3.0 min, 50:50 at 5.0 min, 30:70 at 9.0 min, 5:95 at 10.0 min, 95:5 at 14.0 min; gradient program of negative ion ($v/v$): 95:5 at 0 min, 79:21 at 3.0 min, 50:50 at 5.0 min, 30:70 at 9.0 min, 5:95 at 10.0 min, 95:5 at 14.0 min.

The raw files generated by LC-MS/MS were converted into the mzML format, and ProteoWizard software package (https://proteowizard.sourceforge.io/)[82] was adopted for peak extraction. Alignment and retention time correction were conducted using the R package XCMS program. The peak area was corrected using the support vector regression (SVR) method, and peaks with deletion rate >50% were filtered in each group of samples. A database constructed by Metware Biotechnology was adopted for identification of metabolites. To identify metabolites differentially expressed in WT compared to H2BK17A, three different thresholds were employed: (1) Variable influence on projection (VIP) values ≥1 [based on orthogonal projections to latent structures-discriminant analysis (OPLS-DA) model]; (2) $p < 0.05$ from the two-sided Student's $t$ test, 3) and FC > 2.

## Computational identification of circadian oscillations from omics data

As previously described[34,83], circadian oscillations at different levels were identified by MetaCycle, which incorporates three computational programs including ARSER, JTK_CYCLE and Lomb-Scargle. For our study, the results from ARSER were selected for rhythms detection, under the default threshold of $p < 0.05$.

## Re-construction of the H2BK17pr-centered signaling network

For the analysis of transcriptional regulations, TF motifs of *H. sapiens* were first downloaded from a previously developed database AnimalTFDB 3.0[81], and their orthologs in *D. melanogaster* were generated by reciprocal best hits (RBHs)[71]. Subsequently, the transcriptional regulations between the *D. melanogaster* TFs and 172 PN genes were predicted by the fimo program in MEME Suite 4.10.0[84]. Using $p < 10^{-5}$ as the threshold, the top 10 TFs with the largest number of potential substrates were retained. In addition, from the 118 genes rhythmically regulated by H2BK17pr, 12 additional UPS genes were added to the network. Since 16 TFs were identified to recognize the enriched motifs of the 1368 genes rhythmically bound by H2BK17pr, the regulations between these TFs and substrates in this network regulated by H2BK17pr in a cyclic manner were also investigated. At last, 8 of 16 TFs were predicted to regulate the transcription of at least one of the 15 genes in a cyclic manner. To display the peak binding time interval of H2BK17pr to these genes, we divided the 15 genes into two groups: the Day group (the peak of H2BK17pr binding was between ZT0 and ZT9) and the Night group (the peak of H2BK17pr binding was between ZT12 and ZT21). As a result, 12 genes were assigned to the Day group, while 3 genes were categorized into the Night group. In addition, potential regulators of H2BK17pr were also integrated into the network, including the 3 KATs (NEJ, TIP60 and MOF) and 2 KDACs (HDAC1 and HDAC3).

## Quantification and statistical analysis

The methods of quantification and statistical analysis for propionylomic data analysis, ChIP-seq data analysis, RNA-seq data analysis and metabolomic data analysis were described in corresponding Method Details subsections. Standard two-sided statistical tests used for omic data analysis included Spearman's correlation, hypergeometric test, and Student's $t$ test.

For other experiments, quantifications in all experimental graphs represent the mean of at least three biological replicates, and error bars represent standard deviation (SD). Two-tailed Mann−Whitney U test for unpaired comparisons and two-tailed Student's $t$ test were used to calculate significance. All statistical analysis was carried out using *Statskingdom*. In each statistical analysis, number of biological replicates and significant $p$ values were noted in the figure legends. Asterisks reflecting the calculated $p$ values < 0.05. ns, $p > 0.05$; *$p < 0.05$; **$p < 0.01$; ***$p < 0.001$.

## Reporting summary

Further information on research design is available in the Nature Portfolio Reporting Summary linked to this article.

## Data availability

The raw ChIP-seq and RNA-seq data have been deposited to the National Genomics Data Center (NGDC, https://ngdc.cncb.ac.cn/)[85] with the data set identifier PRJCA007845. The raw MS/MS data of propionylomic and metabolomic profiles have been deposited to the integrated proteome resources (iProX, https://www.iprox.org/)[86]. All annotated MS/MS spectra of propionylomic profiling were also provided. The accession number of the MS/MS data reported in this paper is IPX0003936000, and associated ProteomeXchange accession code is PXD040987. Source data are provided with this paper. Eukaryotic H2B protein sequences were downloaded from the database HistoneDB 2.0 (https://www.ncbi.nlm.nih.gov/research/HistoneDB2.0/index.fcgi/browse/)[36]. The *D. melanogaster* proteome database was obtained from UniProt (https://www.uniprot.org/, Version 201706)[57]. Details of propionyl modification were acquired from the database Unimod (http://www.unimod.org/)[58]. Previously reported Kpr sites used for model training were taken from the database CPLM 4.0 (http://cplm.biocuckoo.cn/)[27]. The predicted 3D structure of fly H2B protein was downloaded from the AlphaFold Protein Structure Database (https://www.alphafold.ebi.ac.uk/)[63] as a PDB file. UPS enzymes/regulators and ALS regulators incorporated in this study were extracted from the database iUUCD (http://iuucd.biocuckoo.org/)[74] and THANANTOS (http://thanatos.biocuckoo.org/)[75], respectively. TFs of *H. sapiens* and *D. melanogaster* and corresponding classifications were downloaded from the database AnimalTFDB 3.0 (http://bioinfo.life.hust.edu.cn/AnimalTFDB/#!/)[81]. Source data are provided with this paper.

## Code availability

The source code of KprFunc has been uploaded to GitHub (https://github.com/CuckooWang/KprFunc) with the DOI identifier (https://doi.org/10.5281/zenodo.4798325)[87].

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

## Acknowledgements

This work was supported by grants from the Natural Science Foundation of China (31930021 and 32022035 to L.Z., 31970633 to Y.X. and 32200956 to K.S.), the Ministry of Science and Technology of China (STI 2030-Major Projects 2021ZD0203200-02 to L.Z., and 2022YFC2704300 and 2021YFF0702000 to Y.X.), Hubei Innovation Group Project (2021CFA005 to Y.X.), and China Postdoctoral Science Foundation (2019T120648 to D.P., 2020M682395 to C.W., and 2021M701337 to W.N.). We thank Dr. Xi Zhou (Wuhan University) for *Drosophila* S2 cells and Dr. Chengqi Xu (HUST) for human AC16 cells used in this study. We thank Drs. Qi Liu (Tongji University) and Li Yang (Fudan University) for their helpful discussion on the small-sample learning technology developed in this study. We also thank Drs. Fangqing Zhao (Beijing Institutes of Life Science, Chinese Academy of Sciences) and Liang Ge (Tsinghua University) for their comments and suggestions regarding this manuscript. We thank Dr. Ying Zhang for her help in data processing.

## Author contributions

Y.X. and L.Z. initiated the project and oversaw all aspects of the project. C.W. developed KprFunc, with the help of W.N. Y.X. and C.W. carried out the data analysis, with the help of W.N., W.Z., M.C., D.P., H.H., A.G., Z.F., and M.Y. K.S., S.M. and Q.L. performed the experiments under the supervision of L.Z. X.Z., and G.G. generated the H2BK17A line. Y.X. and L.Z. wrote the paper with input from all the authors. All authors reviewed and approved the paper for publication.

## Competing interests

The authors declare no competing interests.
