## [Peer Review File · Nature Communications]

Small-sample learning reveals propionylation in determining global protein homeostasisREVIEWER COMMENTS

Reviewer #2 (Remarks to the Author):

Lysine propionylation is a newly discovered post-translational modification, and is increasingly becoming a hot topic due to its critical roles in the cellular process. Although a large volume of effort has been paid to investigate mechanism and function of propionylation, little was known about knowledge of propionylation function. The manuscript proposed a new small-sample learning-based method for lysine propionylation prediction and reveal that propionylation regulated expression of 14.7-16.3% of genes in in the proteostasis network, and further determined global protein level. This is a very interesting conclusion which extremely extend understanding of propionylation. However, the following issues should be explained or solved for better understanding the manuscript.

1. In the introduction, authors should introduce motivation and hypothesis to conduct this research. Authors should also review some important works for computational predicting lysine propionylation, such as PMID: 33967828, PMID: 28763688, and PMID: 29059524, and discuss strength and limitations.
2. Authors should explain advantage of small-sample learning over other learning algorithms in predicting propionylation.
3. To help more readers especially molecular biologists to use the proposed method, authors are encouraged to develop a corresponding webserver or stand-alone software for propionylation prediction.

Reviewer #3 (Remarks to the Author):

In this study, Shui et al. used a mass spectrometry approach to identify *Drosophila* proteins that are modified via propionylation, a post-translational modification on lysine residues. They then developed a machine learning framework to rank propionylated lysines in terms of functional importance, leading them to focus in this study on histone H2B lysine 17 (in humans, this residue is H2BK23). To test the functional relevance of this modification, they developed flies that are mutant for H2BK17A together with an antibody against propionylated H2BK17. Using these reagents, they show that this lysine residue on H2B regulates expression of many genes involved in proteostasis, and suggest that this modification might provide a link between daily feeding cycles and control of proteostasis. Although the data connecting this lysine residue on H2B (K17 *Drosophila*/K23 humans) to proteostasis via gene expression regulation are convincing and very interesting – the authors do not have sufficient controls to conclude that propionylation of this residue rather than another PTM is the relevant modification. Several proteomic studies have shown that *Drosophila* H2BK17/Human H2BK23 can be acetylated or subject to other modification (eg <https://www.science.org/doi/10.1126/scisignal.2001902> , <https://pubs.acs.org/doi/10.1021/ac500972x>), so my major concern is that there are not enough data to conclude that propionylation is the relevant target. It is particularly challenging to distinguish between the functional relevance of acetylation versus propionylation of lysine residues because HATs and HDACs control deposition and removal, respectively, of both of these marks. Based on the propionate versus acetate feeding experiments, I think that propionylation could indeed be the relevant target for H2BK17 regulation of proteostasis, so if the authors can include some additional controls to support this conclusion – in my opinion, this manuscript would be of wide interest to the chromatin community.

Major comments:

1. Most importantly, I recommend further characterization of the antibody developed against propionylated H2BK17 to test if this antibody is specific for propionylated H2BK17 versus acetylated H2BK17 (Extended Data Fig. 3b). Currently the antibody has only been shown to recognize H2BK17Pr,

and since many experiments depend on the specificity of this antibody – it is important to show that it does not recognize acetylated H2BK17 peptides (or human H2BK23ac peptides).

2. In all the experiments in which cells are treated with sodium propionate versus sodium acetate (eg Fig. 3f, 4k), the authors should include controls for other acetylated histone modifications to show that acetate increases levels of these modifications, but propionate does not. I recommend including an antibody against one of the histone H3 acetyl marks, and in the human cell line experiments – including one of the commercially available H2BK23ac antibodies. This would really help to distinguish the impact of acetylation versus propionylation at this H2B residue.

3. The authors have a strong focus on the circadian connection to propionylation and H2BK17Pr in this study that is not entirely supported by the data. Based on the data presented, it looks very much like levels of this mark are regulated simply in response to dietary intake (which in turn is controlled largely by the circadian clock). This is still pretty interesting, but is a bit hard to follow with the way the data is currently presented. If the authors can revise the results text to make this point clearer, that would be helpful.

4. The ChIP-seq data should include a control for nucleosome occupancy such as histone H2B, and I would like to see some IGV examples of gene profiles shown for the H2BK17Pr data. This is an important part of connecting this mark to gene expression regulation, so it would be helpful to move some of these data to a main figure rather than supplemental. It would also be really useful to have an acetyl mark (like an H3 acetyl mark or H2BK23ac) to compare with these – are the distributions similar or distinct?

5. The authors should include some discussion of other PTMs identified at H2BK17/H2BK23.

Minor concerns:

1. Additional details in the figure legends would help with interpretation of these data eg Fig 5e - g, and consistency with figure labeling (eg propionate Fig 4i, sodium propionate Fig 4k, Pr in Fig 6h). Some of the font size is pretty small and a bit hard to read as well.
2. Statistical tests should be described clearly in figure legends, and SDEV rather than SEM would be more appropriate for error bars with relatively small sample sizes.
3. Abbreviations in the text and figure legends are not always consistent, and quite often unnecessary. KM is an odd abbreviation to use for H2BK17A – simply using H2BK17A would be much clearer.

Reviewer #4 (Remarks to the Author):

The authors study the lysine propionylome in *Drosophila* and have developed an interesting machine learning program to pinpoint the Kpr sites that have higher chances to be functionally relevant. They decide to follow H2BK17pr, yet I am not convinced that they correctly validated the antibody developed against this mark, which I am afraid makes the rest of the study questionable.

Results:

Line 126: I cannot find the LP score in Table S1a.

I do not find figures 1b and 1d useful. The spectral counts are obviously usually equal to 1 when analyzing a complex sample, and it is enough to say the content of figure 1d in one line in the text. "Through levels" should be explained (figure 1f).

Line 143: please define per0.

At lines lines 147-148, the authors mention that « the abundance of propionyl-CoA is comparable to acetyl-CoA, thus the complexity of propionylation in proteins may be comparable to acetylation as well". I find this assertion too clear-cut. I have checked in the three mentioned references (21, 22 and

24) and indeed reference 21 says that "propionyl-CoA and butyryl-CoA are present at high concentrations in cells. In the case of starved mouse liver, the concentrations of the two CoAs are only 1–3 times less than acetyl-CoA (15)." I did not find information about a high concentration of propionyl-CoA in reference 22. Finally, reference 24 states that "the stoichiometry of H3K14 acetylation and propionylation are comparable in both HeLa cells and myogenic cells, an interesting observation considering that acetyl-CoA is 10-15-fold more abundant than propionyl-CoA in whole cell measurements [19,25]." In the presently submitted article, I would suggest to be more precise by indicating in which cell types it has been established that propionyl-CoA cellular concentration is (quite) close to that of acetyl-CoA; maybe also explicitly mention the above case of high Kpr stoichiometry. What is the knowledge on relative propionyl-CoA/acetyl-CoA concentrations in *Drosophila*?

I wished to look at the MS data deposited on iProX and saw only RAW files, which I do not have time to run. I would have appreciated easy access to annotated MS/MS spectra. For proteomics data, I need to ask whether the authors tested the specificity of enrichment of Kpr-containing peptides, as the structure of propionylation is close to that of acetylation and both Kpr and Kac peptides are likely enriched by the antibody used. This specificity should be assessed by allowing both Kac and Kpr in the database search. The authors should indicate in the text of the article what fraction of non-modified, acetylated and propionylated peptides were identified in their samples. Beside, did the authors check for the presence of a diagnostic ion at m/z 140.1075 in MS/MS spectra? This fragment is characteristic of the presence of a Kpr in the fragmented peptide. Visual inspection of MS/MS spectra should be done; it would be useful that the authors state in the text whether they indeed observed this low-mass fragment in the vast majority of MS/MS spectra identifying propionylated peptides. A few MS/MS spectra should also be provided as suppl. information. For metabolomics data, the indication of TMT labeling and trypsin digestion on the iProX website is obviously not relevant.

I have gone through table S1 and get lost with the numerous columns, many of which seem to be useless as they only contain empty cells (or "0"). I would recommend that the authors provide simplified tables with only the necessary columns for understanding the results. The antibody developed against H2BK17pr was only validated by dot blot by depositing the target peptide and the corresponding non-modified peptide (Suppl Figure 3b). The peptide modified by H2BK17ac should have been also spotted on the dot blot, as cross-reactivity can be expected against this modification which is very close to propionylation. At line 225, the observation of no signal in the H2BK17A mutant can also come from the impossibility to detect H2BK17ac. The authors seem not to have in mind that this mutation abrogates all types of acylations on that lysine, not only propionylation; the statement at line 226 is incorrect. When analyzing by LC-MS/MS the proteins fished with this antibody, did the authors test for the possible presence of the peptide containing H2BK17ac? All in all, I am not convinced that the present work specifically follows H2BK17pr and not H2BK17ac. It would be good to check what functions have been attributed to H2BK17ac in *Drosophila* and compare them to the current observations tentatively attributed to H2BK17pr.

Material and methods:

Line 598: please define "DD"

Line 611: sonification needs to be corrected into sonication

Lines 631: how the 60 fractions were reduced to 8 fractions needs to be explained

Line 635: it is not the lot number but the reference that is provided

Line 641: Solvent A must be described

Line 658: the full name of CPLM should be given here; are the 1129 non-redundant sites from fly only?

Line 670: Was the SOD2K132pro indeed not in CPLM, while it has been functionally characterized?

Line 677: please correct "was" into "were"

Line 681: please correct "missed cleavages"

Lines 717-720: please explicitly define PWL, SMO, PLR.

Line 728: what is the BLOUSM62 matrix?

Generation of H2BK17pr antibody: what are the two chosen sequences? Please localize them into H2B protein sequences.

Line 879: how was homogenization performed?

Line 882: what is the link between the 10^7 cells and the formerly mentioned 150 mg of fly heads?

Line 884: 20 uL out of what total volume?

Line 886: IgG from which animal as compared to the animal used to produce the antibody of interest?

Line 898: what volume of trypsin (to know the amount used)?

Line 902: please specify the stationary phase of the column: diameter, porosity, material branched.

Line 909: which UPLC was used?

Lines 1020: please say clearly when you get nuclei in the protocol.

Line 1029: which sonicator was used?

Line 1034: what amount of antibody or IgG was used?

Line 1036: is the indicated volume for the slurry or packed beads?

Line 1045: what does "CE" stand for?

Line 1061: why did the authors use a dissymmetrical window around the TSS?

Metabolic profiling: I cannot follow the evolution of sample volumes, and more precisely at what point two tubes are being used.

Lines 1092, 1094 and 1107: please change "was" for "were".

Line 1128: was p300 not a possible candidate?

Line 1129: "KADCs" should be changed into "KDACs".

Detailed Responses to Reviewers' Comments

We thank the editor and the reviewers for their thoughtful comments and suggestions which have made the manuscript much stronger. We have addressed these comments and suggestions as described below. The original reviews are listed point-by-point. Our responses are in blue font. Edits made in the text of the manuscript are marked in red.

Reviewer #2:

1. Lysine propionylation is a newly discovered post-translational modification, and is increasingly becoming a hot topic due to its critical roles in the cellular process. Although a large volume of effort has been paid to investigate mechanism and function of propionylation, little was known about knowledge of propionylation function. The manuscript proposed a new small-sample learning-based method for lysine propionylation prediction and reveal that propionylation regulated expression of 14.7-16.3% of genes in in the proteostasis network, and further determined global protein level. This is a very interesting conclusion which extremely extend understanding of propionylation. However, the following issues should be explained or solved for better understanding the manuscript.

Reply: Thank you for your comments. We are glad to see that you find our work interesting.

2. In the introduction, authors should introduce motivation and hypothesis to conduct this research.

Reply: Thank you for this suggestion. We have now carefully revised the introduction section to describe our motivation and hypothesis, which reads as follows (marked in red):

“...Human individuals carrying mutations in genes encoding propionyl-CoA carboxylase or methylmalonyl-CoA mutase are affected with propionic acidemia (PA) and methylmalonic acidemia (MMA), **which are characterized by life-threatening acute metabolic decompensation (AMD) with excessive protein catabolism**¹⁹. These enzymes are responsible for converting propionyl-CoA to succinyl-CoA, and thus their deficiency results in accumulation of propionyl-CoA which in turn leads to increased protein propionylation²⁰. **However, it is not known whether Kpr is involved in regulating proteostasis.**

To investigate the function of Kpr, we conducted propionylomic profiling of *Drosophila* heads and quantified 171 propionylated proteins containing 344 Kpr sites. Due to the varying functions of PTM sites^{21,22}, we hypothesized that only a small subset of Kpr sites might be functionally important in regulating biological processes. To avoid over-fitting during model training, we developed a new small-sample learning framework, prediction of Kpr sites with functional relevance (KprFunc)...”

3. Authors should also review some important works for computational predicting lysine propionylation, such as PMID: 33967828, PMID: 28763688, and PMID: 29059524, and discuss strength and limitations.

Reply: We have now summarized the three published computational analyses on prediction of lysine propionylation in the last paragraph of Supplementary Discussion as shown below:

“Besides KprFunc, three computational methods were previously developed for prediction of general Kpr sites. In 2017, Ju et al. obtained 413 experimentally identified Kpr sites in 192 proteins¹⁹ from our protein lysine modification database (PLMD 3.0)²⁰, an earlier version of the Compendium of Protein Lysine Modifications (CPLM 4.0)²¹. After redundant and homologous clearance, they compiled a benchmark data set of 327 Kpr sites in 164 proteins. Four types of sequence features, including amino acid composition (AAC), amino acid factors (AAF) of various physicochemical properties in the AAIndex database, binary encoding (BE), and composition of *k*-spaced amino acid pairs (CKSAAP), were integrated and the computational model was trained by the biased support vector machine (SVM). Then they developed an online service of PropPred for prediction of Kpr sites, with an area under the curve (AUC) value of 0.7966 from the 10-fold cross-validation¹⁹. In addition, Wang et al. collected 1,782 known and non-redundant Kpr sites in 820 prokaryotic proteins, and developed PropSeek for specifically predicting prokaryotic Kpr sites²². Six types of sequence features were encoded and integrated, while the SVM method was used for model training. The 10-fold cross-validation was performed on various data sets, with AUC values ranging from 0.798 to 0.845. Recently, Li et al. implemented a new strategy of deep learning followed by transfer learning²³. They obtained 9,584 malonylation sites in 3,429 proteins from our PLMD 3.0²⁰, and then used a deep learning framework of recurrent neural network (RNN) to construct a computational model for predicting

malonylation sites. Then, they obtained 408 Kpr sites in 189 proteins from PLMD 3.0 and the literature. The initial RNN model was used for feature extraction from the Kpr sites, and SVM was used to train the final model. Through the 10-fold cross-validation, such a strategy achieved an AUC value of 0.8088, which was comparable to PropPred. The three methods were not developed for the purpose of predicting the functional relevance of Kpr sites. Moreover, the update of CPLM 4.0 renders more Kpr sites available in this study for pre-training a more reliable model to capture the sequence features of Kpr sites.”

4. Authors should explain advantage of small-sample learning over other learning algorithms in predicting propionylation.

Reply: For prediction of the functional relevance of Kpr sites, both the sequence features and functional characteristics of Kpr sites should be unbiasedly learnt. However, directly training a model based on the 13 known functional Kpr sites will learn the functional characteristics of Kpr sites, while the sequence features of Kpr sites cannot be fully learnt. The small-sample learning framework, KprFunc, implemented a pre-training followed by fine-tuning strategy, in which the former will learn the sequence features of Kpr sites using a much larger benchmark data set of 1,707 known Kpr sites, and the latter will fine-tune the initial model to further incorporate the functional characteristics learnt from the 13 known functional Kpr sites. In comparison, we found that directly trained models will have higher or comparative AUC values relative to KprFunc on prediction of the functional characteristics of Kpr sites. However, these models cannot distinguish Kpr sites from non-Kpr sites. KprFunc exhibits high accuracy in recognition of true Kpr sites. We added these new results to the 4th paragraph of “Small-sample learning predicts the functional relevance of Kpr sites” in Results section (marked in red):

“To demonstrate the superiority of pre-training followed by fine-tuning in KprFunc, the 13 Kpr sites known to be functional were directly used for model training. Using the 5-fold cross-validation, AUC values of DNN, SVMs, RF, and KNN algorithm ranged from 0.7607 to 0.9768 (Supplementary Fig. 3e), implying that the functional characteristics of Kpr sites were well captured. However, when using the 1,707 known Kpr sites for testing, KprFunc showed a promising accuracy while other models could not distinguish Kpr sites from non-Kpr sites (Supplementary Fig. 3f), indicating that

these *ab initio* models were highly biased and failed to learn the sequence features of Kpr sites....”

5. To help more readers especially molecular biologists to use the proposed method, authors are encouraged to develop a corresponding webserver or stand-alone software for propionylation prediction.

Rely: Thank you for your suggestion. We have now developed an online service of KprFunc for academic research purposes. We included a description regarding this in the last paragraph of “Small-sample learning predicts the functional relevance of Kpr sites” in Results section (marked in red):

“For convenience, an online service of KprFunc was developed (<http://kprfunc.biocuckoo.cn/>) for prediction of either general or functional Kpr sites. Single or multiple protein sequences in FASTA or eukaryotic linear motif (ELM)³⁴ format can be submitted (Supplementary Fig. 3g). The prediction results will be shown in a tabular list, including protein accession number/ID, propionylation position, residue type, predictor used, flanking peptide, predicted score, and pre-defined cut-off value (Supplementary Fig. 3h).”

We also added description regarding this in Methods section.

“Implementation of the online service

The online service of KprFunc was developed with PHP and JavaScript at <http://kprfunc.biocuckoo.cn/>, in an easy-to-use manner. For prediction of general or functional Kpr sites, three thresholds including “High”, “Medium” and “Low” could be selected based on *Sp* values of 95%, 90% and 85%, respectively. We extensively tested the online service on various web browsers including Microsoft Edge, Google Chrome, and Mozilla Firefox. More details are available at <http://kprfunc.biocuckoo.cn/userguide.php>.”

Reviewer #3:

1. In this study, Shui et al. used a mass spectrometry approach to identify *Drosophila* proteins that are modified via propionylation, a post-translational modification on

lysine residues. They then developed a machine learning framework to rank propionylated lysines in terms of functional importance, leading them to focus in this study on histone H2B lysine 17 (in humans, this residue is H2BK23). To test the functional relevance of this modification, they developed flies that are mutant for H2BK17A together with an antibody against propionylated H2BK17. Using these reagents, they show that this lysine residue on H2B regulates expression of many genes involved in proteostasis, and suggest that this modification might provide a link between daily feeding cycles and control of proteostasis. Although the data connecting this lysine residue on H2B (K17 *Drosophila*/K23 humans) to proteostasis via gene expression regulation are convincing and very interesting – the authors do not have sufficient controls to conclude that propionylation of this residue rather than another PTM is the relevant modification. Several proteomic studies have shown that *Drosophila* H2BK17/Human H2BK23 can be acetylated or subject to other modification (eg <https://www.science.org/doi/10.1126/scisignal.2001902>, <https://pubs.acs.org/doi/10.1021/ac500972x>), so my major concern is that there are not enough data to conclude that propionylation is the relevant target. It is particularly challenging to distinguish between the functional relevance of acetylation versus propionylation of lysine residues because HATs and HDACs control deposition and removal, respectively, of both of these marks. Based on the propionate versus acetate feeding experiments, I think that propionylation could indeed be the relevant target for H2BK17 regulation of proteostasis, so if the authors can include some additional controls to support this conclusion – in my opinion, this manuscript would be of wide interest to the chromatin community.

Reply: Thank you for your interest in our work and your constructive criticisms.

2. Major comments:

Most importantly, I recommend further characterization of the antibody developed against propionylated H2BK17 to test if this antibody is specific for propionylated H2BK17 versus acetylated H2BK17 (Supplementary Fig. 3b). Currently the antibody has only been shown to recognize H2BK17Pr, and since many experiments depend on the specificity of this antibody – it is important to show that it does not recognize acetylated H2BK17 peptides (or human H2BK23ac peptides).

Reply: Thank you for bringing up this important point. We have now conducted this experiment and found that H2BK17pr antibody does not label non-modified peptide or acetylated peptide. This is shown in Supplementary Fig. 5b and relevant description has been added to the beginning of the second paragraph of “H2BK17pr is a conserved Kpr site driven by daytime TRF” in Results (marked in red): “To validate that the H2BK17 site in *Drosophila* is indeed propionylated, we generated an antibody that specifically detects this particular modification **but not acetylated or non-modified peptide** (Supplementary Fig. 5a and 5b).”

3. In all the experiments in which cells are treated with sodium propionate versus sodium acetate (eg Fig. 3f, 4k), the authors should include controls for other acetylated histone modifications to show that acetate increases levels of these modifications, but propionate does not. I recommend including an antibody against one of the histone H3 acetyl marks, and in the human cell line experiments – including one of the commercially available H2BK23ac antibodies. This would really help to distinguish the impact of acetylation versus propionylation at this H2B residue.

Reply: Thank you for the suggestion. We have now conducted this experiment as requested by assessing the effects of acetate and propionate treatment on H3ac, H3K27ac, and H2BK23ac (in mammalian cells). As can be seen in Supplementary Fig 8, propionate does not prominently alter histone acetylation levels. However, we were not able to observe an increase of acetylation in response to acetate treatment. We reason this may be because acetyl-CoA is a major metabolic intermediate and thus its level may be more tightly controlled within the cell and more difficult to perturb as a consequence. Given that propionate treatment can substantially alter H2BK17pr but not acetylation at this site, and that mutating this site in vivo leads to the opposite effect on protein level vs. that of propionate treatment, we think most likely the alterations of proteostasis that we observe here is mediated by propionylation at H2BK17.

4. The authors have a strong focus on the circadian connection to propionylation and H2BK17Pr in this study that is not entirely supported by the data. Based on the data presented, it looks very much like levels of this mark are regulated simply in response to dietary intake (which in turn is controlled largely by the circadian clock). This is still pretty interesting, but is a bit hard to follow with the way the data is currently presented. If the authors can revise the results text to make this point clearer, that would be helpful.

Reply: We apologize for not explaining this point as clearly as we should have in the original manuscript. We completely agree with you that the major driver of the rhythm in H2BK17pr is the feeding cycle, based on the TRF experiments. However, we did notice that if we limit feeding to the night (nighttime TRF), this will result in loss of H2BK17pr rhythm (Supplementary Fig. 6a and 6b). Moreover, daytime TRF enhances H2BK17pr cycling in WT flies, but it cannot drive the rhythm of H2BK17pr in *per⁰* flies (Supplementary Fig. 6c-6e). Therefore, the conclusion that we draw from these results is that although dietary intake possibly increases H2BK17pr, this is only effective during certain periods of the day (perhaps only during the daytime). Therefore, we have now re-written the relevant results and describe this phenomenon as “feeding cycles generate rhythmic signals that lead to cyclic propionylation/depropionylation at H2BK17, while this process may be gated by the circadian clock”. We have changed the subtitle of this section in the Results to “H2BK17pr is a conserved Kpr site driven by daytime TRF”. We have also modified parts of this section as well as Discussion (the third paragraph in Discussion session), all of which are marked in red.

5. The ChIP-seq data should include a control for nucleosome occupancy such as histone H2B, and I would like to see some IGV examples of gene profiles shown for the H2BK17Pr data. This is an important part of connecting this mark to gene expression regulation, so it would be helpful to move some of these data to a main figure rather than supplemental. It would also be really useful to have an acetyl mark (like an H3 acetyl mark or H2BK23ac) to compare with these – are the distributions similar or distinct?

Reply: Thank you for this suggestion. We have now conducted H2B and H3K27ac ChIPseq profiling. We initially intended to use H2BK23ac for this analysis but it cannot detect fly H2BK17ac based on dot blot experiment (data not shown). Very few co-localization can be observed for H2B and H2BK17pr peaks near the TSS, while substantial co-localization was observed for H3K27ac and H2BK17pr (Supplementary Fig. 13). However, H2BK17pr occupancy in the genome is by far more rhythmic than that of H3K27ac, and there is very little correlation between the two in terms of temporal binding pattern (Supplementary Fig. 13d-13f). We have also included a couple more IGV examples to illustrate these key points (Supplementary Fig. 13e). We have moved most of the H2BK17pr ChIP-seq data into main figures (Fig. 5d-5f and 6c) and Results section. We have also added a new section in the Results with the subtitle

“H2BK17pr displays distinctive temporal distribution profile in the genome”. In addition, we also mentioned these new results in the 4th paragraph of Discussion (marked in red): “This is further supported by the observation that most of the genes bound by H2KB17pr are also occupied by the active transcription marker H3K27ac (Supplementary Fig. 13c).”

6. *The authors should include some discussion of other PTMs identified at H2BK17/H2BK23.*

Reply: Thank you for this suggestion. This has been added to the 2nd paragraph of Discussion (marked in red): “It is noteworthy that besides propionylation, acetylation, butyrylation, crotonylation, hydroxyl-butyrylation and di-methylation have also been reported to occur at H2BK23, while the fly H2BK17 site can be acetylated^{17,28,50,51}. Currently, we have not quite ruled out the influences of these other modifications on proteostasis, and an interesting future direction would be to investigate the function of these modifications at H2BK17/H2BK23 and their interactions with propionylation.”

7. *Minor concerns:*

Additional details in the figure legends would help with interpretation of these data eg Fig 5e - g, and consistency with figure labeling (eg propionate Fig 4i, sodium propionate Fig 4k, Pr in Fig 6h). Some of the font size is pretty small and a bit hard to read as well.

Reply: We have re-written the legends for Fig. 5e-g (which are now Fig. 5g-i) with more details. Hopefully they are easier to understand now. We have changed all relevant labels in Fig. 4 and 6 to “PR” and “AC”. We have increased the font size in Fig. 5 and 6.

8. *Statistical tests should be described clearly in figure legends, and SDEV rather than SEM would be more appropriate for error bars with relatively small sample sizes.*

Reply: We have added descriptions of statistical tests to all figure legends applicable. We have also changed SEM to SD.

9. *Abbreviations in the text and figure legends are not always consistent, and quite often unnecessary. KM is an odd abbreviation to use for H2BK17A – simply using H2BK17A would be much clearer.*

Reply: We have changed KM to H2BK17A in all text and figure legends, as well as in all figures.

Reviewer #4:

1. *The authors study the lysine propionylome in Drosophila and have developed an interesting machine learning program to pinpoint the Kpr sites that have higher chances to be functionally relevant. They decide to follow H2BK17pr, yet I am not convinced that they correctly validated the antibody developed against this mark, which I am afraid makes the rest of the study questionable.*

Reply: Thank you for your constructive criticisms. We have tried our best to address your concerns and hopefully this would be satisfactory.

2. *Results:*

Line 126: I cannot find the LP score in Table S1a.

Reply: We have modified Table S1, and the LP scores are now in the 6th column entitled “Localization prob.”.

3. *I do not find figures 1b and 1d useful. The spectral counts are obviously usually equal to 1 when analyzing a complex sample, and it is enough to say the content of figure 1d in one line in the text. “Through levels” should be explained (figure 1f).*

Reply: Thank you for the suggestion. We have now moved Fig. 1b and 1d to Supplementary figures, and have shortened relevant description in the second paragraph of “Dynamic Kpr profiling in fly heads” in Results section, which now reads as “Kpr sites have a LP score of 1, indicating a high reliability of Kpr site identification, and we detected only one Kpr site for the majority of the proteins (100; 58.48%) (Fig. 1b and Supplementary Fig. 1c)”. Also, we apologize for the typo and we have now changed “through levels” to “trough levels”.

4. *Line 143: please define per0.*

Reply: This had been added and reads as follows (marked in red): “Among the 79 Kpr sites that show prominent temporal variation, propionylation level at 59 (~75%) sites are affected by **mutation of clock gene *per* (*per*⁰)** with >1.2 FC”.

5. At lines lines 147-148, the authors mention that « the abundance of propionyl-CoA is comparable to acetyl-CoA, thus the complexity of propionylation in proteins may be comparable to acetylation as well”. I find this assertion too clear-cut. I have checked in the three mentioned references (21, 22 and 24) and indeed reference 21 says that “propionyl-CoA and butyryl-CoA are present at high concentrations in cells. In the case of starved mouse liver, the concentrations of the two CoAs are only 1–3 times less than acetyl-CoA (15).” I did not find information about a high concentration of propionyl-CoA in reference 22. Finally, reference 24 states that “the stoichiometry of H3K14 acetylation and propionylation are comparable in both HeLa cells and myogenic cells, an interesting observation considering that acetyl-CoA is 10-15-fold more abundant than propionyl-CoA in whole cell measurements [19,25].” In the presently submitted article, I would suggest to be more precise by indicating in which cell types it has been established that propionyl-CoA cellular concentration is (quite) close to that of acetyl-CoA; maybe also explicitly mention the above case of high Kpr stoichiometry. What is the knowledge on relative propionyl-CoA/acetyl-CoA concentrations in *Drosophila*?

Reply: Thank you for bringing up this important point. There is no previous knowledge regarding the relative propionyl-CoA/acetyl-CoA concentrations in flies. Therefore, we decided to assess this instead of relying on published mammalian data which may be quite different from the in vivo environment in flies. We employed commercial ELISA kits to measure propionyl-CoA and acetyl-CoA in fly heads, and found no significant difference in abundance (Fig. 1c). We have added a description regarding this in the second paragraph of “Dynamic Kpr profiling in fly heads” in Results section (marked in red): “**Since the number of Kpr sites detected here is considerably fewer than the 2,023 Kac sites reported in flies, we wondered if this could be due to a lower abundance of propionyl-CoA relative to acetyl-CoA^{27,28}. We measured the level of these two co-enzymes in fly heads but observed no significant difference (Fig. 1c). This implies that the complexity of Kpr in proteins may be comparable to Kac.**”

6. I wished to look at the MS data deposited on iProX and saw only RAW files, which I do not have time to run. I would have appreciated easy access to annotated MS/MS spectra.

Reply: All annotated MS/MS spectra can now be downloaded from <https://www.iprox.cn/> with the identifier IPX0003936000.

7. For proteomics data, I need to ask whether the authors tested the specificity of enrichment of Kpr-containing peptides, as the structure of propionylation is close to that of acetylation and both Kpr and Kac peptides are likely enriched by the antibody used. This specificity should be assessed by allowing both Kac and Kpr in the database search. The authors should indicate in the text of the article what fraction of non-modified, acetylated and propionylated peptides were identified in their samples. Beside, did the authors check for the presence of a diagnostic ion at m/z 140.1075 in MS/MS spectra? This fragment is characteristic of the presence of a Kpr in the fragmented peptide. Visual inspection of MS/MS spectra should be done; it would be useful that the authors state in the text whether they indeed observed this low-mass fragment in the vast majority of MS/MS spectra identifying propionylated peptides. A few MS/MS spectra should also be provided as suppl. information.

Reply: Thank you for this critical suggestion. We have now conducted database search against both propionylation and acetylation. This resulted in the identification of 114 non-modified and 488 modified peptides, while the majority (363/488) of modified peptides are propionylated peptides (Supplementary Fig. 1g; Supplementary Table 1c and 1e). In parallel, MS/MS spectra of propionylated peptides were checked manually for the diagnostic ion at m/z 140.1075 for Kpr, and this fragment was observed in all propionylated peptides detected (Supplementary Fig. 2). We have included relevant descriptions in Results (the last paragraph of “Dynamic Kpr profiling in fly heads”, marked in red): “To evaluate the specificity of our Kpr enrichment process, MaxQuant²⁶ was re-used to simultaneously search both Kpr and Kac peptides. This resulted in the identification of 114 non-modified and 488 modified peptides, while the majority (363/488) of modified peptides are propionylated peptides (Supplementary Fig. 1g; Supplementary Table 1c and 1e). Also, we manually checked the MS/MS spectra of propionylated peptides for the diagnostic ion at m/z 140.1075 for Kpr²⁹, and

this fragment was observed in all propionylated peptides detected (Supplementary Fig. 2).”

8. *For metabolomics data, the indication of TMT labeling and trypsin digestion on the iProx website is obviously not relevant.*

Reply: This has been modified. We have updated the corresponding descriptions for each data type.

9. *I have gone through table S1 and get lost with the numerous columns, many of which seem to be useless as they only contain empty cells (or “0”). I would recommend that the authors provide simplified tables with only the necessary columns for understanding the results.*

Reply: We apologize for not presenting these data in a more succinct manner in our original manuscript. Table S1 has now been modified. Only columns with important information were retained.

10. *The antibody developed against H2BK17pr was only validated by dot blot by depositing the target peptide and the corresponding non-modified peptide (Suppl Figure 3b). The peptide modified by H2BK17ac should have been also spotted on the dot blot, as cross-reactivity can be expected against this modification which is very close to propionylation. At line 225, the observation of no signal in the H2BK17A mutant can also come from the impossibility to detect H2BK17ac. The authors seem not to have in mind that this mutation abrogates all types of acylations on that lysine, not only propionylation; the statement at line 226 is incorrect. When analyzing by LC-MS/MS the proteins fished with this antibody, did the authors test for the possible presence of the peptide containing H2BK17ac? All in all, I am not convinced that the present work specifically follows H2BK17pr and not H2BK17ac. It would be good to check what functions have been attributed to H2BK17ac in Drosophila and compare them to the current observations tentatively attributed to H2BK17pr.*

Reply: We have conducted additional dot blot experiment and found that H2BK17pr antibody does not label non-modified peptide or acetylated peptide. This is shown in **Supplementary Fig. 5b** and relevant description has been added to the results section as well. We completely agree with you that other modifications occur at H2BK17, besides propionylation. Indeed, it has been reported that this site is also acetylated in flies

(<https://www.science.org/doi/10.1126/scisignal.2001902>). We agree it is not appropriate to state that lack of H2BK17pr signal in H2BK17A flies verifies the specificity of this antibody, and thus have removed this sentence at Line 226 (the second paragraph in “H2BK17pr is a conserved Kpr site driven by daytime TRF” section). Based on our LC-MS/MS data, this antibody fishes both H2BK17pr and H2BK17ac (Supplementary Fig. 5e), which may reflect some non-specific binding. But considering this antibody clearly cannot label H2BK17ac on dot blot, we suspect another possibility is that the acetylated H2B binds with the propionylated H2B in vivo and thus are pulled down by the H2BK17pr antibody together with propionylated H2B.

Unfortunately, currently nothing is known regarding the function of these modifications at H2BK17/H2BK23. We agree the mutant experiment alone is not sufficient to attribute the **increase of total protein level** to loss of propionylation at H2BK17. However, addition of propionate but not acetate to cells leads to the opposite effect (i.e. **reduction of total protein level**), and this is accompanied by an increase of H2BK17pr/H2BK23pr. We have now included additional results to demonstrate that propionate treatment does not increase H2BK23ac (Supplementary Fig. 8). Taken together, we believe that although other modifications cannot be ruled out, our results suggest that propionylation is the most likely modification to mediate the effects observed on proteostasis. We also acknowledge this and added relevant discussion (the 2nd paragraph of Discussion, marked in red): “It is noteworthy that besides propionylation, acetylation, butyrylation, crotonylation, hydroxyl-butyrylation and dimethylation have also been reported to occur at H2BK23, while the fly H2BK17 site can be acetylated^{17,28,50,51}. Currently, we have not quite ruled out the influences of these other modifications on proteostasis, and an interesting future direction would be to investigate the function of these modifications at H2BK17/H2BK23 and their interactions with propionylation.”

11. Material and methods:

Line 598: please define “DD”

Reply: This has been added and now reads as “**constant darkness (DD)**”.

12. Line 611: sonification needs to be corrected into sonication

Reply: This has been done.

13. Lines 631: *how the 60 fractions were reduced to 8 fractions needs to be explained*

Reply: This has been added and now reads as “**fractions were combined into 8 fractions at 8 intervals**”.

14. Line 635: *it is not the lot number but the reference that is provided*

Reply: This has been fixed and now reads as “**SKU: PTM-202, PTM Bio**”.

15. Line 641: *Solvent A must be described*

Reply: This has been added and now reads as “**solvent A (0.1% formic acid and 2% acetonitrile)**”.

16. Line 658: *the full name of CPLM should be given here; are the 1129 non-redundant sites from fly only?*

Reply: This has been added and now reads as “**from a previously developed database, Compendium of Protein Lysine Modifications (CPLM 4.0)**”. The 1,129 non-redundant sites are from all organisms that have been reported, not just flies.

17. Line 670: *Was the SOD2K132pro indeed not in CPLM, while it has been functionally characterized?*

Reply: We double checked this site in CPLM 4.0. SOD2K132ac but not SOD2K132pr was found. The latter was published after data collection of CPLM 4.0.

18. Line 677: *please correct “was” into “were”*

Reply: This has been done.

19. Line 681: *please correct “missed cleavages”*

Reply: This has been done.

20. Lines 717-720: *please explicitly define PWD, SMO, PLR.*

Reply: The full names of PWD, SMO and PLR were mentioned in the main text as position weight determination, scoring matrix optimization, and penalized logistic regression, respectively. We also included additional description in Methods which now reads as “**...In GPS 5.0³¹, PWD determines the weight of each position in the**

phosphorylated peptide, and SMO optimizes the scoring matrix from an initial amino acid substitution matrix (e.g., BLOSUM62) with a pairwise identity of no more than 62%⁶⁰. The optimized weights and matrix were then used by the final model to calculate average similarity scores between peptides, which results in higher performance in distinguishing phosphorylated vs. non-phosphorylated sites compared to the original model. In this study, such a strategy was not directly used for prediction, but adopted to embed PSP(30, 30) items into 1D vectors, which were then iteratively optimized by PLR, a classical machine learning algorithm, prior to model training.”

21. Line 728: what is the BLOUSM62 matrix?

Reply: The BLOUSM62 matrix is the most widely used amino acid substitution matrix with a pairwise identity of no more than 62% (*Proc Natl Acad Sci U S A*, 1992, 89, 10915-9, PMID: 1438297). In this study, BLOSUM62 was used as the initial matrix for KprFunc. Due to the procedure of iterative optimization, selecting another amino acid substitution matrix does not influence the finally embedded 1D vectors.

22. Generation of H2BK17pr antibody: what are the two chosen sequences? Please localize them into H2B protein sequences.

Reply: This has been added and now reads as “Two peptides, “CAGKAQ-(propionyl)K-NITKTDK” (peptide 1, from Ala12 to Lys24 of H2B) and “KAGKAQ-(propionyl)K-NITKTC” (peptide 2, from Lys11 to Thr22 of H2B) were synthesized and immunized in New Zealand rabbit (PTM BIO).”

23. Line 879: how was homogenization performed?

Reply: We apologize for the mistake. It should be “suspended” and not “homogenized” here. We have fixed this in the first paragraph of Immunoprecipitation and LC-MS/MS analysis, marked in red.

24. Line 882: what is the link between the 10⁷ cells and the formerly mentioned 150 mg of fly heads?

Reply: We apologize for the confusion. 150 mg refers to in vivo experiments while 10⁷ cells refers to the cell culture experiments. We have clarified this in the first paragraph of Immunoprecipitation and LC-MS/MS analysis, marked in red.

25. Line 884: 20 uL out of what total volume?

Reply: This has been added and now reads as “20 µl of 450 µl”.

26. Line 886: IgG from which animal as compared to the animal used to produce the antibody of interest?

Reply: This has been added and now reads as “rabbit normal IgG”.

27. Line 898: what volume of trypsin (to know the amount used)?

Reply: This has been added and now reads as “equal volume of 20 ng/µl trypsin”.

28. Line 902: please specify the stationary phase of the column: diameter, porosity, material branched.

Reply: This has been added and now reads as “packed with 1.9 µm/120 Å ReproSil-PurC18 resins (Dr. Maisch GmbH, Ammerbuch, Germany).”

29. Line 909: which UPLC was used?

Reply: This has been added and now reads as “EASY-nLC 1000 UPLC system (Thermo Fisher Scientific)”.

30. Lines 1020: please say clearly when you get nuclei in the protocol.

Reply: This has been modified (the first paragraph of ChIP-seq profiling and data analysis, marked in red).

31. Line 1029: which sonicator was used?

Reply: This has been added and now reads as “Samples were sonicated with 650W Φ3 sonicator (JY92-IIN, Scientz)”.

32. Line 1034: what amount of antibody or IgG was used?

Reply: This has been added and now reads as “5 µg antibody or normal IgG from same species w”.

33. Line 1036: is the indicated volume for the slurry or packed beads?

Reply: Slurry. This has been edited and now reads as “50 µl of Protein A magnetic bead slurry (Bio-Rad)”.

34. Line 1045: what does “CE” stand for?

Reply: ChIP Elution. This has been fixed.

35. Line 1061: why did the authors use a dissymmetrical window around the TSS?

Reply: Based on previous work (e.g., *Nature*, 2015, 518, 344-9, PMID: 25693565), this window enriches promoters where potential transcription factor (TF) usually binds and is a commonly selected window for ChIP-seq studies focusing on transcriptional regulation. We have rephrased the corresponding description which now reads as “...H2BK17pr binding signals were searched from sequences -2000 to 500 bp surrounding TSSs of genes to annotate peaks at promoters⁷⁸.”

36. Metabolic profiling: I cannot follow the evolution of sample volumes, and more precisely at what point two tubes are being used.

Reply: We apologize for the confusion and have re-written this part which now reads as “In order to increase extraction efficiency, we used methanol/water and acetate/methanol to extract metabolites sequentially. 50 mg of fly heads per sample were homogenized with 500 µl ice-cold methanol/water (70%, v/v). Homogenates were vortexed for 5 min and incubated on ice for 15 min, followed by centrifugation at 12,000 rpm at 4 °C for 10 min. 400 µl supernatant was collected into tube 1. 500 µl ethyl acetate/methanol (1:3, v/v) was added into the original centrifuge tube, and the mixture was oscillated for 5 min and incubated on ice for 15 min. The mixture was centrifuged at 12,000 rpm at 4 °C for 10 min and 400 µl of supernatant was collected into tube 2. Tube 1 and 2 were combined and the 800 µl supernatants were vacuum dried.”

37. Lines 1092, 1094 and 1107: please change “was” for “were”.

Reply: This has been changed.

38. Line 1128: was p300 not a possible candidate?

Reply: We apologize for the confusion. NEJ is the fly homologue of mammalian p300 and CBP.

39. *Line 1129: “KADCs” should be changed into “KDACs”.*

Reply: This has been changed.

REVIEWERS' COMMENTS

Reviewer #3 (Remarks to the Author):

The authors have addressed my concerns regarding the specificity of their H2BK17pr antibody and the cross-talk between acetylation and propionylation in the acetate/propionate supplementation experiments. The discussion in the text about the rhythmic H2BK17pr levels in relation to dietary intake/circadian clock control is also much clearer than in the previous version. I have no additional concerns.

Reviewer #4 (Remarks to the Author):

I thank the authors for re-processing their MS/MS data while including Kac as a variable modification. It is satisfying to see that Kpr-containing peptides seem to represent the majority of identifications. Regarding the detection of the diagnostic ion at 140.1075, I acknowledge that it should not always be very intense (it should be intense when Kpr is the first amino acid of the fragmented peptide), but I must say I am not convinced by the two MS/MS spectra shown in Suppl Figure 2. First, it is a pity the resolution of the two images showing the spectra is so poor. In spectrum b, even though it is hard to read masses, it does not seem that the zoom was made around the fragment at 140.1 Da. Second, I am puzzled to observe that the higher-intensity fragments detected in these two spectra, especially at higher masses for spectrum b, do not correspond to fragments matching the proposed peptide sequence. The authors should really be able to show more convincing spectra identifying Kpr-containing peptides.

I did not find any data on iProX when providing the ID IPX0003936000 in the "Search" item. I could find 4 datasets from the last author of this article, but not the one of interest here.

MS/MS spectra shown in Figure 3i and j need to be of better resolution, so that a reader can easily look at the masses and labels of fragments. Fragment y7 is not well placed in spectrum 3j, it must be at mass 900.55 and thus needs to be pushed to the left. If a signal is detected at 140.1075 in spectrum 3i (it seems to me it could be the case), it should really be labelled as "diag(Kpr)", as it would support the proposed peptide identification. Again, in these two MS/MS spectra, I am bothered by the fact that some higher intensity fragments are not interpreted in terms of theoretical fragments expected for the suggested sequences. Could the central base peak (of highest intensity) correspond to residual non-fragmented precursor ion? The m/z values of precursor peptides should be provided in the legend. Could the authors check whether two peptides were co-selected for fragmentation, which would explain the production and detection of fragments coming from two sequences?

Minor points:

Line 724, "fractions were combined into 8 fractions at 8 intervals" is still not clear to me.

Line 1019, the expression "equal volume of 20 ng/μl" still does not tell the reader how much trypsin was applied onto each gel band. This is not a critical information, but the only precise information is to tell the amount (in ug) of trypsin applied, not a volume nor a concentration.

There must be typing mistakes in "H3BK27ac" at lines 479-480 .

Detailed Responses to Reviewers' Comments

We thank the editor and the reviewers for their thoughtful comments and suggestions which have made the manuscript much stronger. We have addressed these comments and suggestions as described below. The original reviews are listed point-by-point. Our responses are in blue font. Edits made in the text of the manuscript are marked in red.

Reviewer #3:

1. The authors have addressed my concerns regarding the specificity of their H2BK17pr antibody and the cross-talk between acetylation and propionylation in the acetate/propionate supplementation experiments. The discussion in the text about the rhythmic H2BK17pr levels in relation to dietary intake/circadian clock control is also much clearer than in the previous version. I have no additional concerns.

Reply: Thank you for your comments.

Reviewer #4:

1. I thank the authors for re-processing their MS/MS data while including Kac as a variable modification. It is satisfying to see that Kpr-containing peptides seem to represent the majority of identifications.

Reply: Thank you, we are glad to know that our data are satisfying.

2. Regarding the detection of the diagnostic ion at 140.1075, I acknowledge that it should not always be very intense (it should be intense when Kpr is the first amino acid of the fragmented peptide), but I must say I am not convinced by the two MS/MS spectra shown in Suppl Figure 2. First, it is a pity the resolution of the two images showing the spectra is so poor. In spectrum b, even though it is hard to read masses, it does not seem that the zoom was made around the fragment at 140.1 Da.

Reply: To address your concerns, we re-annotated the two MS/MS spectra shown in Supplementary Fig. 2, using a program developed in house which is adapted from Glyco-Decipher, a tool for analysis of glycopeptide spectra (*Nat Commun*, 2022, 13, 1900; PMID: 35393418). This analysis was performed with the help from Dr.

Mingliang Ye and his student, Zheng Fang. In this revised version, the *b1* ion was detected in addition in both S2a and S2b.

3. *Second, I am puzzled to observe that the higher-intensity fragments detected in these two spectra, especially at higher masses for spectrum b, do not correspond to fragments matching the proposed peptide sequence. The authors should really be able to show more convincing spectra identifying Kpr-containing peptides.*

Reply: We calculated the *m/z* values of precursor peptides for both MS/MS spectra in Supplementary Fig. 2, and the peak with the highest intensity for each spectrum tends to be derived from the divalent precursor peptide.

4. *I did not find any data on iProX when providing the ID IPX0003936000 in the “Search” item. I could find 4 datasets from the last author of this article, but not the one of interest here.*

Reply: We apologize for this inconvenience. In this revised version, we have released all raw data on iProX with the accession number IPX0003936000.

5. *MS/MS spectra shown in Figure 3i and j need to be of better resolution, so that a reader can easily look at the masses and labels of fragments. Fragment y7 is not well placed in spectrum 3j, it must be at mass 900.55 and thus needs to be pushed to the left.*

Reply: We have re-generated Figure 3i and 3j which is now of higher resolution, and re-labeled all fragment ions. In this revised version, we detected the *b6* ion in addition in 3i, and *b4* and *b6* ions in 3j.

6. *If a signal is detected at 140.1075 in spectrum 3i (it seems to me it could be the case), it should really be labelled as “diag(Kpr)”, as it would support the proposed peptide identification.*

Reply: Thank you for the suggestion. This has been done.

7. *Again, in these two MS/MS spectra, I am bothered by the fact that some higher intensity fragments are not interpreted in terms of theoretical fragments expected for the suggested sequences. Could the central base peak (of highest intensity) correspond to residual non-fragmented precursor ion?*

Reply: We have tried our best to annotate the MS/MS spectra. In Fig. 3j and Supplementary Fig. 2a and b, the base peak of highest intensity corresponds to the m/z value of the divalent precursor peptide, indicating that the base peak was derived from the residual non-fragmented precursor ion. In Fig. 3i, we detected the divalent precursor ion, although it is not the base peak of highest intensity.

8. *The m/z values of precursor peptides should be provided in the legend. Could the authors check whether two peptides were co-selected for fragmentation, which would explain the production and detection of fragments coming from two sequences?*

Reply: The m/z values of precursor peptides were provided for both Fig. 3i and 3j. For Fig. 3i, we manually checked adjacent isolation windows of its precursor ion during MS/MS data acquisition, and did not observe any co-fragmented peptides around the isolation window of the precursor ion. This means the unlabeled peak of highest intensity in Fig. 3i is not a result of co-fragmented peptides. Peptides were identified by MaxQuant search under a stringent threshold with false discovery rates of <1% for PSM, protein and Kpr site decoy fraction. Moreover, the continuous $y1$ - $y6$ ions and the divalent precursor ion at 658.9235 help reflect the reliability of the AQK(pr)NITK peptide, together with our series of further experimental analysis of this site.

9. *Minor points:*

Line 724, “fractions were combined into 8 fractions at 8 intervals” is still not clear to me.

Reply: We apologize for not having described this more clearly. We now added an additional Supplementary Data 1e to provide the details for the compositions of the 8 fractions.

10. *Line 1019, the expression ““equal volume of 20 ng/ μ l” still does not tell the reader how much trypsin was applied onto each gel band. This is not a critical information, but the only precise information is to tell the amount (in ug) of trypsin applied, not a volume nor a concentration.*

Reply: This has been added and now reads as “200 μ l of 20 ng/ μ l”.

11. *There must be typing mistakes in “H3BK27ac” at lines 479-480.*

Repy: Thank you for pointing this out. We have edited it.